# On the Geometry of Regularization in Adversarial Training: High-Dimensional Asymptotics and Generalization Bounds

**Matteo Vilucchio**                                              *matteo.vilucchio@epfl.ch*
*Information Learning and Physics Laboratory, École Polytechnique Fédérale de Lausanne (EPFL)*

**Nikolaos Tsilivis**                                              *nt2231@nyu.edu*
*Center for Data Science, New York University*

**Bruno Loureiro**                                              *brloureiro@gmail.com*
*Département d'Informatique, École Normale Supérieure - PSL & CNRS, France*

**Julia Kempe**                                              *kempe@nyu.edu*
*Center for Data Science, New York University*
*Courant Institute of Mathematical Sciences, New York University*
*FAIR, META*

**Reviewed on OpenReview:** *https://openreview.net/forum?id=vkmvuranbm*

## Abstract

Regularization, whether explicit in terms of a penalty in the loss or implicit in the choice of algorithm, is a cornerstone of modern machine learning. Indeed, controlling the complexity of the model class is particularly important when data is scarce, noisy or contaminated, as it translates a statistical belief on the underlying structure of the data. This work investigates the question of how to choose the regularization norm $\|\cdot\|$ in the context of high-dimensional adversarial training for binary classification. To this end, we first derive an exact asymptotic description of the robust, regularized empirical risk minimizer for various types of adversarial attacks and regularization norms (including non-$\ell_p$ norms). We complement this analysis with a uniform convergence analysis, deriving bounds on the Rademacher Complexity for this class of problems. Leveraging our theoretical results, we quantitatively characterize the relationship between perturbation size and the optimal choice of $\|\cdot\|$, confirming the intuition that, in the data scarce regime, the type of regularization becomes increasingly important for adversarial training as perturbations grow in size.

## 1 Introduction

Despite all its successes, deep learning still underperforms spectacularly in worst-case situations, when models face innocent-looking data which are adversarially crafted for eliciting erroneous or undesired outputs. Since the discovery of these failure modes in computer vision (Szegedy et al., 2014) and their re-discovery, more recently, in other modalities including text (Zou et al., 2023), considerable effort has been put in designing algorithms for training models which are robust against these adversarial attacks.

In the context of supervised learning problems, a principled approach consists of appropriately modifying standard empirical risk minimization: a parametric model is fit by minimizing a *worst-case* empirical risk, where "worst-case" refers to an assumed threat model. For example, in computer vision, a threat model of $\ell_\infty$ perturbations translates the assumption that images whose pixels only differ by a little should share the same label. Despite its conceptual clarity and proven ability to return robust models, a major drawback of this method, known as *robust empirical risk minimization* (RERM) or adversarial training (Goodfellow et al., 2015a; Madry et al., 2018), is that it often comes with a performance tradeoff, besides being computationally more intensive than standard ERM. Indeed, it has been observed that model accuracy is often compromised

for better robustness (Tsipras et al., 2019; Zhang et al., 2019). To make matters worse, neural networks often exhibit a large gap between their robust train and test performances in standard computer vision benchmarks (Rice et al., 2020).

Many empirical efforts in addressing these statistical limitations of RERM have focused on either increasing the amount of labeled (Wang et al., 2023) or unlabeled (Carmon et al., 2019; Zhai et al., 2019) data, or on painstakingly re-imagining several of the design choices of deep learning (such at the loss function (Zhai et al., 2019), model averaging (Chen et al., 2021; Rebuffi et al., 2021) and more). Despite the apparent empirical challenges, simple guidelines on how different choices affect the statistical efficiency of RERM are clearly missing, even in simple models.

In this work, we make a step towards theoretically filling this gap by investigating model selection in RERM, and how it relates to robust and standard generalization error. In particular, we focus on the oldest model selection method: (weight) *regularization*. Following a large body of work originating in high-dimensional statistics (Krogh & Hertz, 1991; Seung et al., 1992; Bean et al., 2013a; Thrampoulidis et al., 2018; Aubin et al., 2020; Vilucchio et al., 2024; 2025), we study this fundamental question *asymptotically*, when both the input dimension and the number of training samples grow to infinity while keeping their ratio constant, and under a Gaussian setting. While it is customary in this literature to study which *values* of regularization coefficients yield the best test errors (balancing empirical fitness with model complexity), we instead analyze the optimality of a *type* of regularization. A motivation for this comes from a separate line of work in uniform convergence bounds that stresses the importance of the type of regularization for robust generalization (Yin et al., 2019; Awasthi et al., 2020; Tsilivis et al., 2024). Borrowing from this line of work, which mainly offers qualitative bounds, and reinforcing it with new findings, we demonstrate, via *sharp* asymptotic descriptions of the errors in (regularized) RERM for a variety of different perturbation and regularization norms, that regularization becomes increasingly important in RERM as the perturbation strength grows in size. This allows us to get an exact description of the relationship between optimal type of regularization and strength of perturbation, and discuss how regularization affects the tradeoff between robustness and accuracy.

To summarize, our **main contributions** in this work are the following:

1. We derive an exact asymptotic description of the performance of regularized RERMs for a variety of perturbation and regularization norms. In addition to the usually studied $\ell_p$, we consider $\|\cdot\|_\Sigma$ norms (induced by a positive symmetric matrix $\Sigma$), which allow us to separate the effect of a perturbation on different features of the input.

2. We show uniform convergence bounds for this class of problems (i.e., $\|\cdot\|_\Sigma$ regularized), by establishing new results on the Rademacher complexities for several classes of linear hypothesis classes under adversarial perturbations.

3. Leveraging the theoretical results above, we show that regularizing with the *dual* norm of the perturbation can yield benefits in terms of robustness and accuracy, compared to other regularization choices. In particular, our analysis permits a precise characterization of the relationship between the perturbation geometry and the optimal type of regularization. It further allows a decomposition of the contribution of regularization in terms of standard and robust (test) error.

Our results can be seen as positive news. Indeed, the main implication of our work for robust machine learning practice is that model selection, in the form of either explicit or implicit regularization, plays a more important role in robust ERM than in standard ERM. In the context of robust deep learning practice, model selection is often implicit in the choice of architecture, learning algorithm, stopping time, hyperparameters, etc. Our theoretical analysis in the context of simple adversarial tasks highlights the importance of these choices, as they can be crucial to the outcome in terms of robustness and performance.

Finally, while typical-case and worst-case analyses usually appear as opposites in the statistical learning literature, we believe our work nicely illustrates how these two approaches to studying generalization can be combined in a complementary way to yield precise answers with both explanatory and predictive powers.

### 1.1 Related work

We discuss here two recent related works, while we defer a more extensive discussion to Appendix A. Recently, Tanner et al. (2025) derived high dimensional asymptotics for robust binary classification with $\ell_2$ regularization, considering perturbations in a general $\|\cdot\|_\Sigma$ norm. In our work, we study the effect of regularization, providing exact asymptotics for any combination of $\ell_p$ perturbation and regularization norm, while extending (Tanner et al., 2025) for various $\|\cdot\|_A$ regularization norms (where $A$ is a positive symmetric matrix). From the perspective of learning theory, Tsilivis et al. (2024) recently highlighted the importance of the (implicit) regularization in RERM with linear models, by showing the effect of the learning algorithm and the architecture on the robustness of the final predictor. In our work, we consider, instead, explicit regularization and more general perturbation (and regularization) geometries.

In Section 3 we present the technical theorems about the exact asymptotics. In Section 4 we present the uniform convergence bounds. In Section 5, using the technical tools developed in the previous two sections, we investigate what regularisation geometry is optimal for robust generalization error.

## 2 Setting Specification

We consider a binary classification task with training data $\mathcal{S} = \{(\boldsymbol{x}_i, y_i)\}_{i=1}^n$, where $\boldsymbol{x}_i \in \mathbb{R}^d$ and $y_i \in \{-1, +1\}$ are sampled independently from a distribution $\mathcal{D}$ of the following form:

$$P(\boldsymbol{x}, y) = \int_{\mathbb{R}^d} \mathrm{d}\boldsymbol{w}_\star \mathbb{P}_{\mathrm{out}}\left(y \Big| \frac{\langle \boldsymbol{w}_\star, \boldsymbol{x} \rangle}{\sqrt{d}}\right) P_{\mathrm{in}}(\boldsymbol{x}) P_{\boldsymbol{w}}(\boldsymbol{w}_\star), \tag{1}$$

where $P_{\mathrm{in}}$ is a probability density function over $\mathbb{R}^d$ and $\mathbb{P}_{\mathrm{out}} : \mathbb{R} \to [0, 1]$ encodes our assumption that the label is a (potentially non-deterministic) linear function of the input $\boldsymbol{x}$ with teacher weights $\boldsymbol{w}_\star \in \mathbb{R}^d$. Here, we denote $z = \langle \boldsymbol{w}_\star, \boldsymbol{x} \rangle / \sqrt{d}$ as the (normalized) pre-activation. For example, a noiseless problem corresponds to $\mathbb{P}_{\mathrm{out}}(y|z) = \delta(y - z)$, while we can incorporate noise by using the *probit* model: $\mathbb{P}_{\mathrm{out}}(y|z) = 1/2 \operatorname{erfc}\left(-yz/\sqrt{2}\tau\right)$, where $\tau > 0$ controls the label noise. We assume that $\boldsymbol{w}_\star \in \mathbb{R}^d$ is drawn from a prior distribution $P_{\mathrm{w}}$.

Given the training data $\mathcal{S}$, our objective is to investigate the robustness and accuracy of linear classifiers $\hat{y}(\hat{\boldsymbol{w}}, \boldsymbol{x}) = \mathrm{sign}(\langle \hat{\boldsymbol{w}}, \boldsymbol{x} \rangle / \sqrt{d})$, where $\hat{\boldsymbol{w}} = \hat{\boldsymbol{w}}(\mathcal{S})$ are learned from the training data.

We define the *robust generalization error* as

$$E_{\mathrm{rob}}(\hat{\boldsymbol{w}}) = \mathbb{E}_{(\boldsymbol{x}, y) \sim \mathcal{D}}\left[\max_{\|\boldsymbol{\delta}\| \leq \varepsilon} \mathbb{1}(y \neq \hat{y}(\hat{\boldsymbol{w}}, \boldsymbol{x} + \boldsymbol{\delta}))\right], \tag{2}$$

where the pair $(\boldsymbol{x}, y)$ comes from the same distribution as the training data, and $\varepsilon$ bounds the magnitude of adversarial perturbations under a specific choice of norm. The (standard) *generalization error* is defined as the rate of misclassification of the learnt predictor

$$E_{\mathrm{gen}}(\hat{\boldsymbol{w}}) = \mathbb{E}_{(\boldsymbol{x}, y) \sim \mathcal{D}}[\mathbb{1}(y \neq \hat{y}(\hat{\boldsymbol{w}}, \boldsymbol{x}))]. \tag{3}$$

Notice that for $\varepsilon = 0$: $E_{\mathrm{rob}}(\hat{\boldsymbol{w}}) = E_{\mathrm{gen}}(\hat{\boldsymbol{w}})$ for all $\hat{\boldsymbol{w}} \in \mathbb{R}^d$. We will frequently use the following decomposition of the robust generalization error into (standard) generalization error and *boundary error* $E_{\mathrm{bnd}}$:

$$E_{\mathrm{rob}}(\hat{\boldsymbol{w}}) = E_{\mathrm{gen}}(\hat{\boldsymbol{w}}) + E_{\mathrm{bnd}}(\hat{\boldsymbol{w}}), \tag{4}$$

where $E_{\mathrm{bnd}}$ is defined as follows

$$E_{\mathrm{bnd}}(\hat{\boldsymbol{w}}) = \mathbb{E}_{(\boldsymbol{x}, y) \sim \mathcal{D}}\left[\mathbb{1}(y = \hat{y}(\hat{\boldsymbol{w}}; \boldsymbol{x})) \max_{\|\boldsymbol{\delta}\| \leq \varepsilon} \mathbb{1}(y \neq \hat{y}(\hat{\boldsymbol{w}}, \boldsymbol{x} + \boldsymbol{\delta}))\right]. \tag{5}$$

As its name suggests, $E_{\mathrm{bnd}}$ is the probability of a sample lying on (or near) the decision boundary, i.e., the probability that a sample is correctly classified without perturbation but incorrectly classified with it.

## 2.1 Robust Regularized Empirical Risk Minimization

Direct minimization of the robust generalization error of eq. (2) presents two main challenges: first, the objective function is non-convex due to the indicator function, and second, we only have access to a finite dataset rather than the full data-generating distribution. To address these issues, a widely adopted approach, introduced for the first time by Goodfellow et al. (2015b), is to optimise the robust *empirical* (regularized) risk, defined as

$$\mathcal{L}(\boldsymbol{w}) = \sum_{i=1}^{n} \max_{\|\boldsymbol{\delta}_i\| \leq \varepsilon} g\left(y_i \frac{\langle \boldsymbol{w}, \boldsymbol{x}_i + \boldsymbol{\delta}_i \rangle}{\sqrt{d}}\right) + \lambda \widetilde{r}(\boldsymbol{w}), \tag{6}$$

where $g : \mathbb{R} \to \mathbb{R}_+$ is a non-increasing convex loss function that serves as a surrogate for the 0-1 loss, $\widetilde{r}(\boldsymbol{w})$ is a convex regularization term, and $\lambda \geq 0$ is a regularization parameter. The inner maximization over $\boldsymbol{\delta}_i$ models the worst-case perturbation for each data point, constrained by the attack budget $\varepsilon$ during training. Given the dataset $\mathcal{S}$, we estimate the parameters of our model as

$$\hat{\boldsymbol{w}} \in \arg\min_{\boldsymbol{w} \in \mathbb{R}^d} \mathcal{L}(\boldsymbol{w}). \tag{7}$$

The choice of loss function $g$, regularization $\widetilde{r}$, and parameters $\varepsilon$ and $\lambda$ can significantly impact the model's accuracy and robustness.

In practice, eq. (7) is often solved with a first-order optimization method, such as gradient descent. Prior work (Soudry et al., 2018) has shown that optimizing the *unregularized* loss without any adversarial perturbations (eq. (7) for $\lambda, \varepsilon = 0$) with gradient descent is equivalent to eq. (7) with the euclidean norm squared as a regularizer, where the free regularizer strength $\lambda$ corresponds to the time duration of the algorithm ($\lambda \to 0$ as the number of iterations goes to $\infty$). Similar results can be obtained for different first-order algorithms (Gunasekar et al., 2018) (in particular, when $\widetilde{r}(\boldsymbol{w}) = \|\boldsymbol{w}\|_p^p$, this corresponds to the family of *steepest descent* algorithms) as well as in the adversarial case ($\varepsilon > 0$) (Tsilivis et al., 2024). Therefore, studying eqs. (6) and (7) is equivalent to studying the solutions returned by a first-order optimization algorithm.

## 3 Exact Asymptotics of Robust ERM

Our first technical result is an asymptotic description of the properties of the solution of eqs. (6) and (7) in the proportional high-dimensional limit, under the assumption of isotropic Gaussian distribution. While restrictive, this assumption is supported by recent theoretical advances showing that many learning problems exhibit universality: their asymptotic behavior matches Gaussian predictions even with non-Gaussian data (Goldt et al., 2022b; Loureiro et al., 2021; Hu & Lu, 2023; Montanari & Saeed, 2022; Dandi et al., 2023; Wei et al., 2022; Pesce et al., 2023; Gerace et al., 2024). While proving such correspondence for our setting is outside the scope of this work, this suggests our analysis of the Gaussian case can provide valuable insights into practical adversarial training.

### 3.1 Results for $\ell_p$ norms

First, we consider the setting where the perturbations in eqs. (2) and (6) are constrained in their $\ell_p$ norm for $p \in (1, \infty]$. More precisely, we make the following assumptions:

**Assumption 3.1** (High-Dimensional Limit). We consider the proportional high-dimensional regime where both the number of training data $n$ and input dimension $d$ diverge to infinity simultaneously at the same rate, while maintaining a fixed ratio $\alpha := n/d$.

**Assumption 3.2** ($\ell_p$ Norms). Let $\|\boldsymbol{x}\|_p = (\sum_{i=1}^{n} |x_i|^p)^{1/p}$ denote the $\ell_p$ norm for $p \in (1, \infty]$, with $p^\star$ being its dual exponent ($1/p + 1/p^\star = 1$). The adversarial perturbations are constrained by an $\ell_p$ norm with parameter $p$, while for regularization we consider the function $\widetilde{r}(\boldsymbol{w}) = \|\boldsymbol{w}\|_r^r$ where $r \in [1, \infty)$ is a parameter that can differ from $p$.

**Assumption 3.3** (Scaling of Adversarial Norm Constraint). We suppose that the value of $\varepsilon$ scales with the dimension $d$ such that $\varepsilon/(\sqrt{d} \sqrt[p^\star]{d}) = O_d(1)$.

**Assumption 3.4** (Data Distribution). For each $i \in [n]$, the covariates $\boldsymbol{x}_i \in \mathbb{R}^d$ are drawn i.i.d. from the data distribution $P_{\mathrm{in}}(\boldsymbol{x}) = \mathcal{N}_{\boldsymbol{x}}(\mathbf{0}, \mathrm{Id}_d)$. Then the corresponding $y_i$ is sampled independently from the conditional distribution $\mathbb{P}_{\mathrm{out}}$ defined in eq. (1). The target weight vector $\boldsymbol{w}_\star \in \mathbb{R}^d$ is drawn from a prior probability distribution $P_{\boldsymbol{w}}$ which is separable, i.e. $P_{\boldsymbol{w}}(\boldsymbol{w}) = \prod_{i=1}^d P_w(w_i)$ for a distribution $P_w$ in $\mathbb{R}$ with finite variance $\mathrm{Var}(P_w) = \rho < \infty$.

Under these assumptions, our first result states that in the high-dimensional limit, the robust generalization error associated with the RERM solution in eq. (7) asymptotically depends only on a few deterministic variables, known as the *summary statistics*, which can be computed by solving a set of low-dimensional self-consistent equations.

**Theorem 3.5** (Limiting errors for $\ell_p$ norm). *Let $\hat{\boldsymbol{w}}(\mathcal{S}) \in \mathbb{R}^d$ denote a solution of the RERM problem in eq. (7). Then, under Assumptions 3.1 to 3.4, the standard, robust and boundary generalization error of $\hat{\boldsymbol{w}}$ converge in distribution to the following deterministic quantities*

$$E_{\mathrm{gen}}(\hat{\boldsymbol{w}}) = \frac{1}{\pi} \arccos\left( m^\star / \sqrt{(\rho + \tau^2) q^\star} \right),$$

$$E_{\mathrm{bnd}}(\hat{\boldsymbol{w}}) = \int_0^{\varepsilon \frac{p^\star \sqrt{P^\star}}{\sqrt{q^\star}}} \mathrm{erfc}\left( \frac{-\frac{m^\star}{\sqrt{q^\star}}\nu}{\sqrt{2(\rho + \tau^2 - m^{\star 2}/q^\star)}} \right) \frac{e^{-\frac{\nu^2}{2}}}{\sqrt{2\pi}} \, \mathrm{d}\nu,$$

$$E_{\mathrm{rob}}(\hat{\boldsymbol{w}}) = E_{\mathrm{gen}}(\hat{\boldsymbol{w}}) + E_{\mathrm{bnd}}(\hat{\boldsymbol{w}})$$

*where $m^\star, q^\star, P^\star$ are the values to which the following* summary statistics *converge in probability to, i.e.*

$$\frac{1}{d}\langle \boldsymbol{w}_\star, \hat{\boldsymbol{w}} \rangle \to m^\star, \quad \frac{1}{d}\|\hat{\boldsymbol{w}}\|_2^2 \to q^\star, \quad \frac{1}{d}\|\hat{\boldsymbol{w}}\|_{p^\star}^{p^\star} \to P^\star,$$

*Remark* 3.6. An immediate observation from the above equations is that $E_{\mathrm{gen}}$ is monotonically *increasing* as a function of the cosine of the angle between teacher and student ($m^\star/\sqrt{\rho q^\star}$), while $E_{\mathrm{bnd}}$ is *decreasing*. This has been observed before for boundary based classifiers (Tanay & Griffin, 2016; Tanner et al., 2025).

Theorem 3.5 therefore states that in order to characterize the robust generalization error in the high-dimensional limit, it is enough to compute three low-dimensional statistics of the RERM solution. Our next result shows that these quantities can be asymptotically computed without having to actually solve the high-dimensional minimization problem in eq. (7).

**Theorem 3.7** (Self-consistent equations for $\ell_p$ norms). *Under the same assumptions as Theorem 3.5, the summary statistics $(m^\star, q^\star, P^*)$ are the unique solution of the following set of **self-consistent** equations:*

$$\begin{cases} \hat{m} = \alpha \mathbb{E}_\xi \left[ \int_{\mathbb{R}} \mathrm{d}y \, \partial_\omega \mathcal{Z}_0 f_g(\sqrt{q}\xi, P) \right] \\ \hat{q} = \alpha \mathbb{E}_\xi \left[ \int_{\mathbb{R}} \mathrm{d}y \, \mathcal{Z}_0 f_g^2(\sqrt{q}\xi, P) \right] \\ \hat{V} = -\alpha \mathbb{E}_\xi \left[ \int_{\mathbb{R}} \mathrm{d}y \, \mathcal{Z}_0 \partial_\omega f_g(\sqrt{q}\xi, P) \right] \\ \hat{P} = \varepsilon \alpha p^\star P^{-\frac{1}{p}} \mathbb{E}_\xi \left[ \int_{\mathbb{R}} \mathrm{d}y \, \mathcal{Z}_0 y f_g(\sqrt{q}\xi, P) \right] \end{cases}, \qquad \begin{cases} m = \mathbb{E}_\xi \left[ \partial_\gamma \mathcal{Z}_w f_w(\sqrt{\hat{q}}\xi, \hat{P}, \lambda) \right] \\ q = \mathbb{E}_\xi \left[ \mathcal{Z}_w f_w(\sqrt{\hat{q}}\xi, \hat{P}, \lambda)^2 \right] \\ V = \mathbb{E}_\xi \left[ \mathcal{Z}_w \partial_\gamma f_w(\sqrt{\hat{q}}\xi, \hat{P}, \lambda) \right] \\ P = \mathbb{E}_\xi \left[ \mathcal{Z}_w \partial_{\hat{P}} \mathcal{M}_{\frac{\lambda}{\hat{V}}|\cdot|^r + \frac{\hat{P}}{\hat{V}}|\cdot|^{p^\star}} (\frac{\sqrt{\hat{q}}\xi}{\hat{V}}) \right] \end{cases}, \quad (9)$$

$$(8)$$

*where $\mathcal{Z}_w = \mathcal{Z}_w(\hat{m}\xi/\sqrt{\hat{q}}, \hat{m}/\sqrt{\hat{q}})$, $\mathcal{Z}_0 = \mathcal{Z}_0(y, m\xi/\sqrt{q}, \rho - m^2/q)$ and $\xi \sim \mathcal{N}(0,1)$, and:*

$$\mathcal{Z}_0(y, \omega, V) = \mathbb{E}_{z \sim \mathcal{N}(0,1)} \left[ P_{\mathrm{out}}(y \mid \sqrt{V}z + \omega) \right], \qquad \mathcal{Z}_w(\gamma, \Lambda) = \mathbb{E}_{w \sim P_w} \left[ e^{-\frac{1}{2}\Lambda w^2 + \gamma w} \right], \qquad (10)$$

$$f_g(\omega, \hat{P}) = \left( \mathcal{P}_{V g(y, \cdot -y\varepsilon \, p^\star \sqrt{P})}(\omega) - \omega \right)/V, \qquad f_w(\gamma, \hat{P}, \Lambda) = \mathcal{P}_{\frac{\lambda}{\Lambda}|\cdot|^r + \frac{\hat{P}}{\Lambda}|\cdot|^{p^\star}}\left( \frac{\gamma}{\Lambda} \right). \qquad (11)$$

*where we indicate the proximal of a function $f : \mathbb{R} \to \mathbb{R}$ as $\mathcal{P}_{V f(\cdot)}(\omega)$ and its Moreau envelope with $\mathcal{M}_{V f(\cdot)}(\omega)$.*

Two remarks on these two results are in order.

*Remark* 3.8. Both results hold for any separable convex regularizer in the definition of the empirical risk in eq. (6). This is in contrast to many prior works in this field, which primarily consider $\ell_2$ regularizations.

*Remark* 3.9. The first four equations (eq. (8)) depend only on the noise distribution and the loss function, while the second set (eq. (9)) depends on the regularization function and the dual norm of the perturbation.

### 3.2 Results for Mahalanobis norms

While the $\ell_p$ norm is the most frequently discussed in the robust learning literature, $\ell_p$ perturbations are isotropic, treating all covariates equally. Under the isotropic Gaussian Assumption 3.4, this is justified. However, it can be limiting under more realistic scenarios where the covariates are *structured*, and for instance some features are more relevant than others. Recently, Tanner et al. (2025) introduced a model for studying adversarial training under structured covariates which considers perturbations under a *Mahalanobis* norm, allowing to weight the perturbation along different directions. However, the discussion in that work focused only on $\ell_2$ regularization.

Since our goal in this work is to study what is the best regularization choice for a given perturbation geometry, we now derive asymptotic results akin to the ones of Section 3.1 under any combination of Mahalanobis perturbation and regularization norm. As before, we start by introducing our assumptions.

**Assumption 3.10** (Mahalanobis norms). Given a positive definite matrix $\boldsymbol{\Sigma_\delta}$, we consider perturbations under a Mahalanobis norm $\|\boldsymbol{x}\|_{\boldsymbol{\Sigma_\delta}} = \sqrt{\boldsymbol{x}^\top \boldsymbol{\Sigma_\delta} \boldsymbol{x}}$. Additionally, we consider the regularization function to be $\widetilde{r}(\boldsymbol{w}) = {}^1\!/\!2 \ \boldsymbol{w}^\top \boldsymbol{\Sigma_w} \boldsymbol{w}$ for a positive definite matrix $\boldsymbol{\Sigma_w}$.

**Assumption 3.11** (Structured data). For each $i \in [n]$, the covariates $\boldsymbol{x}_i \in \mathbb{R}^d$ are drawn i.i.d. from the data distribution $P_{\text{in}}(\boldsymbol{x}) = \mathcal{N}_{\boldsymbol{x}}(\boldsymbol{0}, \boldsymbol{\Sigma_x})$. Then the corresponding $y_i$ is sampled independently from the conditional distribution $\mathbb{P}_{\text{out}}$ defined in eq. (1). The target weight vector $\boldsymbol{w}_\star \in \mathbb{R}^d$ is drawn from a prior probability distribution $\boldsymbol{w}_\star \sim P_{\boldsymbol{w}} = \mathcal{N}_{\boldsymbol{w}_\star}(\boldsymbol{0}, \boldsymbol{\Sigma_\theta})$, which we assume has limiting Mahalanobis norm given by $\rho = \lim_{d \to \infty} \mathbb{E}[\frac{1}{d} \boldsymbol{w}_\star^\top \boldsymbol{\Sigma_x} \boldsymbol{w}_\star]$.

**Assumption 3.12** (Scaling of Adversarial Norm Constraint). The value of $\varepsilon$ does not scales with the dimension $d$ such that $\varepsilon = O_d(1)$.

**Assumption 3.13** (Convergence of spectra). We suppose that $\boldsymbol{\Sigma_x}, \boldsymbol{\Sigma_\delta}, \boldsymbol{\Sigma_\theta}, \boldsymbol{\Sigma_w}$ are simultaneously diagonalisable. We call $\boldsymbol{\Sigma_x} = \mathrm{S}^\top \mathrm{diag}(\omega_i) \mathrm{S}$, $\boldsymbol{\Sigma_\delta} = \mathrm{S}^\top \mathrm{diag}(\zeta_i) \mathrm{S}$ and $\boldsymbol{\Sigma_w} = \mathrm{S}^\top \mathrm{diag}(w_i) \mathrm{S}$. We define $\bar{\boldsymbol{\theta}} = \mathrm{S} \boldsymbol{\Sigma_x}^\top \boldsymbol{w}_\star / \sqrt{\rho}$. We assume that the empirical distributions of eigenvalues and the entries of $\bar{\boldsymbol{\theta}}$ jointly converge to a probability distribution $\mu$ as

$$\sum_{i=1}^d \delta(\bar{\boldsymbol{\theta}}_i - \bar{\theta}) \delta(\omega_i - \omega) \delta(\zeta_i - \zeta) \delta(w_i - w) \to \mu \,. \tag{12}$$

*Remark* 3.14. The simultaneous diagonalizability assumption is equivalent to the matrices having *common principal components* (Flury, 1984; 1988), a common framework in multivariate statistics where multiple covariance matrices share eigenvectors but may have group-specific eigenvalues. This structure arises naturally in signal processing applications such as blind source separation (Belouchrani et al., 1997; Cardoso & Souloumiac, 1996) and independent component analysis (Comon, 1994; Hyvärinen & Oja, 2000).

As in Section 3.1, our first result concerns the limiting robust error.

**Theorem 3.15** (Limiting errors for Mahalanobis norm). *Let $\hat{\boldsymbol{w}}(\mathcal{S}) \in \mathbb{R}^d$ denote the unique solution of the RERM problem in eq. (7). Then, under Assumptions 3.1 and 3.10 to 3.13, the standard, robust and boundary generalization error of $\hat{\boldsymbol{w}}$ converge in distribution to the following deterministic quantities*

$$E_{\text{gen}}(\hat{\boldsymbol{w}}) = \frac{1}{\pi} \arccos\left( m^\star / \sqrt{(\rho + \tau^2) q^\star} \right),$$

$$E_{\text{bnd}}(\hat{\boldsymbol{w}}) = \int_0^{\varepsilon_g \frac{\sqrt{P^\star}}{\sqrt{q^\star}}} \mathrm{erfc}\left( \frac{-\frac{m^\star}{\sqrt{q^\star}} \nu}{\sqrt{2(\rho + \tau^2 - m^{\star 2}/q^\star)}} \right) \frac{e^{-\frac{\nu^2}{2}}}{\sqrt{2\pi}} \, d\nu,$$

$$E_{\text{rob}}(\hat{\boldsymbol{w}}) = E_{\text{gen}}(\hat{\boldsymbol{w}}) + E_{\text{bnd}}(\hat{\boldsymbol{w}})$$

*where $m^\star, q^\star, P^\star$ are the values to which the following* summary statistics *converge in probability to, i.e.*

$$\frac{\boldsymbol{w}_\star^\top \boldsymbol{\Sigma_x} \hat{\boldsymbol{w}}}{d} \to m^\star \,, \quad \frac{\hat{\boldsymbol{w}}^\top \boldsymbol{\Sigma_x} \hat{\boldsymbol{w}}}{d} \to q^\star \,, \quad \frac{\hat{\boldsymbol{w}}^\top \boldsymbol{\Sigma_\delta} \hat{\boldsymbol{w}}}{d} \to P^\star \,,$$

As in Section 3.1, our next result shows that the summary statistics characterizing the limiting errors can be obtained from of a set of self-consistent equations.

**Theorem 3.16** (Self-Consistent equations for Mahalanobis norm). *Under the same assumptions as Theorem 3.15, the summary statistics $(m^\star, q^\star, P^*)$ are the unique solution of the following set of **self-consistent** equations:*

$$\begin{cases} \hat{m} = \alpha \mathbb{E}_\xi \left[ \int_\mathbb{R} dy \, \partial_\omega \mathcal{Z}_0(y, \sqrt{\eta}\xi, 1-\eta) f_g(\sqrt{q}\xi, P) \right] \\ \hat{q} = \alpha \mathbb{E}_\xi \left[ \int_\mathbb{R} dy \, \mathcal{Z}_0(y, \sqrt{\eta}\xi, 1-\eta) f_g^2(\sqrt{q}\xi, P) \right] \\ \hat{V} = -\alpha \mathbb{E}_\xi \left[ \int_\mathbb{R} dy \, \mathcal{Z}_0(y, \sqrt{\eta}\xi, 1-\eta) \partial_\omega f_g(\sqrt{q}\xi, P) \right] \\ \hat{P} = 2\varepsilon \alpha P^{-\frac{1}{2}} \mathbb{E}_\xi \left[ \int_\mathbb{R} dy \, \mathcal{Z}_0(y, \sqrt{\eta}\xi, 1-\eta) y f_g(\sqrt{q}\xi, P) \right] \end{cases}, \ (13) \qquad \begin{cases} m = \mathbb{E}_\mu \left[ \frac{\hat{m}\bar{\theta}^2}{\lambda w + \hat{V}\omega + \hat{P}\delta} \right] \\ q = \mathbb{E}_\mu \left[ \frac{\hat{m}^2\bar{\theta}^2\omega + \hat{q}\omega^2}{(\lambda w + \hat{V}\omega + \hat{P}\delta)^2} \right] \\ V = \mathbb{E}_\mu \left[ \frac{\omega}{\lambda w + \hat{V}\omega + \hat{P}\delta} \right] \\ P = \mathbb{E}_\mu \left[ \zeta \frac{\hat{m}^2\bar{\theta}^2 + \hat{q}\omega^2}{(\lambda w + \hat{V}\omega + \hat{P}\delta)^2} \right] \end{cases}, \quad (14)$$

*where $\mu$ is the joint limiting distribution for the spectrum of all the matrices from Assumption 3.13.*

*Remark* 3.17. Notice that the first set of equations is the same as in Theorem 3.7, as they depend only on the marginal distribution $\mathbb{P}_{\text{out}}$ and the loss function.

*Remark* 3.18 (Interpretation of the self-consistent equations). The self-consistent equations in Theorems 3.7 and 3.16 reduce the high-dimensional optimization problem in eq. (7) to tracking only three scalar statistics $(m^\star, q^\star, P^\star)$. These equations, derived via Gordon's Min-Max theorem, fully characterize the asymptotic behavior of regularized RERM through the interaction of these quantities at optimality.

While the self-consistent equations in Theorem 3.7 and Theorem 3.16 do not admit a closed-form solution, they can be efficiently solved using an iteration scheme (Appendix E). Solving them yields precise curves for the generalization errors of the final predictor as a function of the sample complexity $\alpha$ and regularization geometry, allowing us to draw conclusions for the interplay between the regularization and perturbation – see simulations in Section 5.

The details of the proofs of Theorems 3.5, 3.7, 3.15 and 3.16 are discussed in Appendix B. They are based on an adaptation of Gordon's Min-Max Theorem for convex empirical risk minimization problems (Thrampoulidis et al., 2014; Loureiro et al., 2021).

## 4 Which Regularization to Choose?

Our results in the previous section provide tight predictions on the robust and standard generalization error of the set of minimizers of the robust (regularized) empirical risk. However, since the self-consistent equations describing the robust errors are not closed, it is not straightforward to read *why* some regularizers might produce better results than others. In this section, we derive complementary uniform convergence bounds based on the *Rademacher Complexity* for linear predictors under various geometries. While these bounds might not be numerically tight, they are distribution-agnostic, and provide a-priori guarantees for the error of a predictor which are *qualitatively* useful. We start by introducing concepts in a general way, before deriving guarantees for the case considered in Section 3.2.

Let $\mathcal{H}_{\widetilde{r}}$ be a hypothesis class of linear predictors of restricted complexity, as captured by a function $\widetilde{r} : \mathbb{R}^d \to \mathbb{R}$. This function $\widetilde{r}$ plays the role of a regularizer, as in Section 3. We define:

$$\mathcal{H}_{\widetilde{r}} = \{\mathbf{x} \to \langle \boldsymbol{w}, \mathbf{x} \rangle : \widetilde{r}(\boldsymbol{w}) \leq \mathcal{W}_{\widetilde{r}}^2\}, \tag{15}$$

where $\mathcal{W}_{\widetilde{r}} > 0$ is an arbitrary upper bound.

Central to the analysis of the generalization error uniformly inside the hypothesis class $\mathcal{H}_{\widetilde{r}}$ is the notion of the (empirical) *Rademacher Complexity* (Koltchinskii, 2001) of $\mathcal{H}_{\widetilde{r}}$:

$$\hat{\mathfrak{R}}_{\text{S}}(\mathcal{H}_{\widetilde{r}}) = \mathbb{E}_\sigma \left[ \frac{1}{n} \sup_{\boldsymbol{w}:\widetilde{r}(\boldsymbol{w}) \leq \mathcal{W}_{\widetilde{r}}^2} \sum_{i=1}^n \sigma_i \langle \boldsymbol{w}, \boldsymbol{x}_i \rangle \right], \tag{16}$$

where the $\sigma_i$'s are either $-1$ or $1$ with equal probability. In the case of robust generalization with respect to $\|\cdot\|$-limited perturbations, it suffices to analyse the *worst-case* Rademacher Complexity of $\mathcal{H}_{\widetilde{r}}$:

$$\hat{\mathfrak{R}}_{\mathrm{S}}(\widetilde{\mathcal{H}}_{\widetilde{r}}) = \mathbb{E}_\sigma \left[ \frac{1}{n} \sup_{\boldsymbol{w}:\widetilde{r}(\boldsymbol{w}) \leq \mathcal{W}_r^2} \sum_{i=1}^n \sigma_i \min_{\|\boldsymbol{\delta}_i\| \leq \varepsilon} \langle \boldsymbol{w}, \boldsymbol{x}_i + \boldsymbol{\delta}_i \rangle \right].$$

With these ingredients in place, we can state the following bound on the robust generalization gap of any predictor in $\mathcal{H}_{\widetilde{r}}$.

**Theorem 4.1** (Mohri et al. (2012); Awasthi et al. (2020))**.** *For any $\delta > 0$, with probability at least $1 - \delta$ over the draw of the dataset $\mathcal{S}$, for all $\boldsymbol{w} \in \mathbb{R}^d$ such that $\widetilde{r}(\boldsymbol{w}) \leq \mathcal{W}_r^2$ (eq. (15)), it holds that*

$$E_{\mathrm{rob}}(\boldsymbol{w}) \leq \hat{E}_{\mathrm{rob}}(\boldsymbol{w}) + 2\,\hat{\mathfrak{R}}_{\mathrm{S}}(\widetilde{\mathcal{H}}_r) + 3\sqrt{\frac{\log 2/\delta}{2n}}, \tag{17}$$

*where*

$$\hat{E}_{\mathrm{rob}}(\boldsymbol{w}) = \frac{1}{n} \sum_{i=1}^n \max_{\|\boldsymbol{\delta}_i\| \leq \varepsilon} \mathbb{1}(y_i \hat{y}(\boldsymbol{w}, \boldsymbol{x}_i + \boldsymbol{\delta}_i) \leq 1) \tag{18}$$

*is a robust empirical error.*

Theorem 4.1 promises that a tight bound on the worst-case Rademacher complexity of $\mathcal{H}_{\widetilde{r}}$ can bound the (robust) generalization gap of any predictor in $\mathcal{H}_{\widetilde{r}}$. The next proposition realises this goal for the general class of *strongly convex* functions $\widetilde{r}$. This will permit the study of the cases of Section 3.2.

**Proposition 4.2.** *Let $\varepsilon, \sigma > 0$. Consider a sample $\mathcal{S} = \{(\boldsymbol{x}_1, y_1), \ldots, (\boldsymbol{x}_n, y_n)\}$, and let $\mathcal{H}_{\widetilde{r}}$ be the hypothesis class defined in eq. (15), where $\widetilde{r}$ is $\sigma$- strongly convex with respect to a norm $_r\|\cdot\|$. Then, it holds:*

$$\hat{\mathfrak{R}}_{\mathrm{S}}(\widetilde{\mathcal{H}}_{\widetilde{r}}) \leq \max_{i \in [n]} {}_r\|\mathbf{x}_i\|_\star \mathcal{W}_{\widetilde{r}} \sqrt{\frac{2}{\sigma n}} + \frac{\varepsilon}{2\sqrt{n}} \sup_{\boldsymbol{w}:\widetilde{r}(\boldsymbol{w}) \leq \mathcal{W}_{\widetilde{r}}^2} \|\boldsymbol{w}\|_\star, \tag{19}$$

*where $_r\|\cdot\|_\star, \|\cdot\|_\star$ denote the dual norms of $_r\|\cdot\|, \|\cdot\|$, respectively.*

The proof (Appendix C) leverages a fundamental result on (standard) Rademacher complexities for strongly convex functions due to Kakade et al. (2008) and a symmetrization argument. This result informs us that the worst-case Rademacher complexity can decompose into two terms: one which characterizes the standard error and one that scales with the magnitude of perturbation $\varepsilon$ and depends on the *dual* norm of the perturbation. Thus, we expect that a regularization which promotes a small second term in the RHS of eq. (19) will likely mean a smaller robust generalization gap, as $\varepsilon$ increases. This can be further elucidated in the following subcases (proofs appear in Appendix C), for which we already derived exact asymptotics in Section 3.2:

- $\|\cdot\| = \|\cdot\|_{\boldsymbol{\Sigma_\delta}}$ and $\widetilde{r}(\boldsymbol{w}) = \|\boldsymbol{w}\|_2^2$: this corresponds to perturbations with respect to a a symmetric positive definite matrix $\boldsymbol{\Sigma_\delta} \in \mathbb{R}^{d \times d}$, while we regularize in the Euclidean norm. In this case, we obtain:

  **Corollary 4.3.** *Let $\varepsilon > 0$ and symmetric positive definite $\boldsymbol{\Sigma_\delta} \in \mathbb{R}^{d \times d}$. Then:*

  $$\hat{\mathfrak{R}}_{\mathrm{S}}(\widetilde{\mathcal{H}}_{\|\cdot\|_2^2}) \leq \frac{\max_{i \in [n]} \|\mathbf{x}_i\|_2 \mathcal{W}_2}{\sqrt{n}} + \frac{\varepsilon \mathcal{W}_2}{2\sqrt{n}} \sqrt{\lambda_{\min}^{-1}(\boldsymbol{\Sigma_\delta})}.$$

- $\|\cdot\| = \|\cdot\|_{\boldsymbol{\Sigma_\delta}}$ and $\widetilde{r}(\boldsymbol{w}) = \|\boldsymbol{w}\|_{\boldsymbol{\Sigma_w}}^2$: this corresponds to perturbations with respect to a symmetric positive definite matrix $\boldsymbol{\Sigma_\delta} \in \mathbb{R}^{d \times d}$ and regularization with respect to a norm induced by another matrix $\boldsymbol{\Sigma_w} \in \mathbb{R}^{d \times d}$. We will analyze the special case where $\boldsymbol{\Sigma_\delta}$ and $\boldsymbol{\Sigma_w}$ share the same set of eigenvectors.

  **Corollary 4.4.** *Let $\varepsilon > 0$. Let $\boldsymbol{\Sigma_w} = \sum_{i=1}^d \alpha_i \mathbf{v}_i \mathbf{v}_i^T$ and $\boldsymbol{\Sigma_\delta} = \sum_{i=1}^d \lambda_i \mathbf{v}_i \mathbf{v}_i^T$, with $\mathbf{v}_i \in \mathbb{R}^d$ being orthonormal. Then:*

  $$\hat{\mathfrak{R}}_{\mathrm{S}}(\widetilde{\mathcal{H}}_{\|\cdot\|_A^2}) \leq \frac{\mathcal{W}_A \max_{i \in [n]} \|\mathbf{x}_i\|_{\boldsymbol{\Sigma_w^{-1}}}}{\sqrt{n}} + \frac{\varepsilon \mathcal{W}_{\boldsymbol{\Sigma_w}}}{2\sqrt{n}} \sqrt{\max_{i \in [d]} \frac{1}{\lambda_i \alpha_i}}. \tag{20}$$

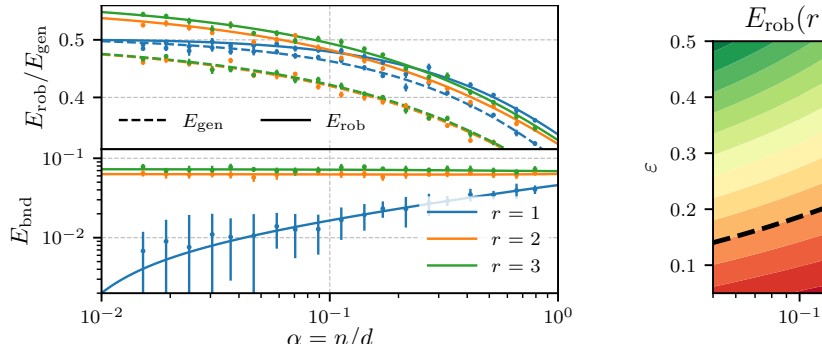
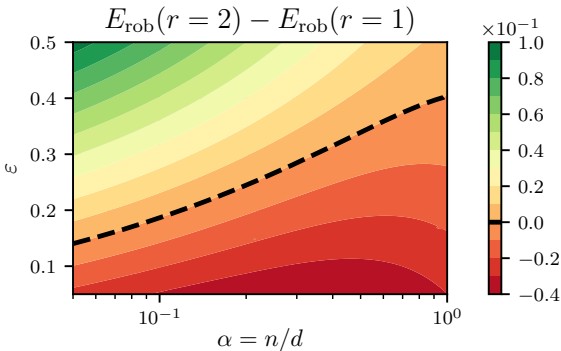

Figure 1: **(Left)** Generalization error of RERMs in the low sample complexity regime under $\ell_\infty$ perturbations for various choices of regularization. The edge of $\ell_1$ over other methods stems from the boundary error ($E_{\mathrm{bnd}}$) which goes to zero as $\alpha \to 0^+$. Setting: $\varepsilon = 0.2$ with optimally tuned $\lambda$. Bullet points with error bars are RERM simulations for $d = 1000$ (10 seeds). **(Right)** Difference between robust generalization errors for $r = 2$ and $r = 1$ as a function of $\varepsilon$ and $\alpha$ for $\ell_\infty$ attacks. Green zones correspond to areas where the dual norm regularization ($\ell_1$) performs better than $\ell_2$.

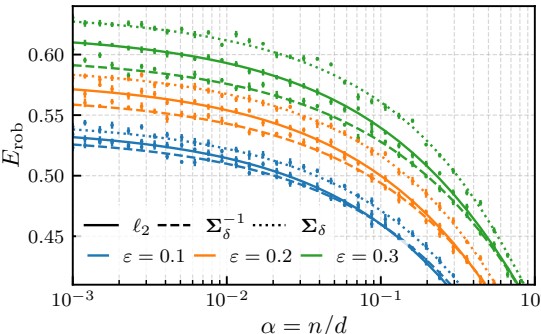
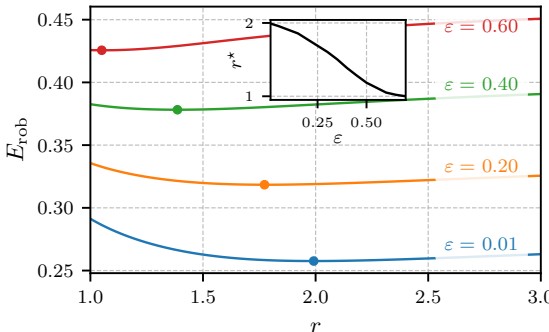

Figure 2: **(Left)** Difference between robust generalization error for $\boldsymbol{\Sigma_\delta}$ perturbations. We see that a regularization with the dual norm has the best adversarial error for different choices of $\varepsilon$. The points with the error bars (std) are RERM simulations for $d = 1000$ (10 random seeds), while the solid lines correspond to the theoretical predictions. **(Right)** Robust generalization error of the solution of regularized RERM as a function of the regularization order $r$, i.e. $\widetilde{r}(\boldsymbol{w}) = \lambda \|\boldsymbol{w}\|_r^r$ for various perturbations strengths $\varepsilon$. Sample complexity $\alpha = 1.0$. Regularization coefficients $\lambda$ are optimally tuned. The inside figure shows how the optimal value of $r$ scales with $\varepsilon$.

Hence, we deduce that regularizing the class of linear predictors with $\boldsymbol{\Sigma_w} = \boldsymbol{\Sigma_\delta}^{-1}$, where $\boldsymbol{\Sigma_\delta}$ is the matrix of the perturbation norm, can more effectively control the robust generalization error. See Remark C.4 for more details.

Similar results can be derived in the context of $\ell_p$ perturbations - see Yin et al. (2019); Awasthi et al. (2020) and Appendix C. In fact, mirroring our analysis, the robust generalization error there is controlled by the $\|\cdot\|_{p^\star}$ norm of the weights. We explore the effect of the regularizer numerically with simulations next.

## 5 Experiments

Leveraging our exact results from Section 3 and guided by the predictions of Section 4, in this Section we numerically investigate the role of the regularization geometry in the robustness and accuracy of robust empirical risk minimizers. Experimental details and further ablation studies are presented in Appendix D, where we consider experiments on the MNIST dataset (Deng, 2012) and additional ablations for choice of $\boldsymbol{\Sigma_w}$ in the Mahalanobis norms' case.

### 5.1 Importance of Regularization in the Scarce Data Regime

First, we consider the setting of Section 3.1, with perturbations constrained in their $\ell_\infty$ norms, for three different regularizers ($\ell_1, \ell_2$ and $\ell_3$ norms). Figure 1 **(Left)** compares the generalization errors of the solutions in eq. (7) for the various regularizers and plots them as a function of the sample complexity $\alpha$. Note that when $\alpha$ is small (scarce data), the $\ell_1$ regularized solution (dual norm of $\ell_\infty$) provides better defense against $\ell_\infty$ perturbations. Interestingly, this is due to the fact that the boundary error approaches zero as $\alpha \to 0^+$, only in the case when $r = 1$ (same figure, bottom). We analytically explore this phenomenon further in Appendix B.8, where we analyze the boundary error from Theorem 3.5 and probe its dependence on the overlap parameters $(m^\star, q^\star, P^\star)$.

The phase diagram of Figure 1 **(Right)** further elucidates the difference of the methods as a function of $\varepsilon$. We display the difference in robust generalization error between $\ell_2$ and $\ell_1$ regularized solutions versus attack budget $\varepsilon$ and sample complexity $\alpha$, with *optimally tuned* regularization coefficient $\lambda$. We observe that $\ell_1$ outperforms $\ell_2$ regularization in regions of high $\varepsilon$ and low $\alpha$.

Figure 2 **(Left)** demonstrates $E_{\mathrm{rob}}$ in the structured case of Section 3.2, where the perturbations are constrained in a Mahalanobis norm $\|\cdot\|_{\boldsymbol{\Sigma}_{\boldsymbol{\delta}}}$. We observe that regularizing the weights of the solution with the dual norm of the perturbation ($\|\cdot\|_{\boldsymbol{\Sigma}_{\boldsymbol{\delta}}^{-1}}$) yields better robustness, while the gap between the various methods increases as $\varepsilon$ grows.

### 5.2 Optimal Regularization Geometry as a Function of $\varepsilon$

While the previous figures compared the various regularizers $\widetilde{r}(\boldsymbol{w})$ as $\varepsilon$ grows, it is not clear what exactly the relationship is between optimal $\widetilde{r}(\boldsymbol{w})$ and perturbation strength. In particular, we expect when $\varepsilon = 0$, $\ell_2$-regularized solutions to achieve better accuracy, due to the fact that the data are Gaussian. However, it is not clear how the transition to the dual norm happens.

We examine this relationship in Figure 2 **(Right)**, where we plot the robust generalization error for various values of perturbation $\varepsilon$ and regularization order $r$ for a fixed value of sample complexity $\alpha$. We observe that, as the attack strength increases, the order of the optimal regularization *smoothly* transitions from $r = 2$ to $r = 1$. Hence, there is a regime of perturbation scale $\varepsilon$ where neither $r = 2$ nor $r = 1$ is optimal, but an order of $r \in (1, 2)$ achieves the least robust test error.

## 6 Conclusion

We studied the role of regularization in robust empirical risk minimization (adversarial training) for a variety of perturbation and regularization norms. We derived an exact asymptotic description of the robust and standard generalization error in the high-dimensional proportional limit, and we showed results for the (worst-case) Rademacher Complexity of linear predictors in the case of structured perturbations. Phase diagrams and exact scaling laws, afforded by our analysis, suggest that choosing the right regularization becomes increasingly important as $\varepsilon$ grows, and, in fact, this optimal regularization often corresponds to the dual norm of the perturbation. Furthermore, our results reveal a curious, smooth, transition between different optimal regularizations ($\ell_2$ to $\ell_1$) with increasing perturbation strength; a phenomenon that has not yet been captured by any other theoretical work.

It would be interesting for future works to investigate the interplay between regularization and perturbation geometry in non-linear models, such as the random features model (Mei & Montanari, 2022; Gerace et al., 2021; Hassani & Javanmard, 2024).

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

# A   Further related Work

**Exact asymptotics:**    Our results on the exact asymptotics of adversarial training build on an established body of literature that spans high-dimensional probability (Thrampoulidis et al., 2014; 2015; Taheri et al., 2023), statistical physics (Mignacco et al., 2020; Gerace et al., 2021; Bordelon et al., 2020; Loureiro et al., 2021; Okajima et al., 2023; Adomaityte et al., 2024; 2023; Tabanelli et al., 2025) and random matrix theory (Bean et al., 2013b; Mai et al., 2019; Liao et al., 2020; Mei & Montanari, 2022; Xiao et al., 2022; Schröder et al., 2023). Our study is also motivated by recent efforts to understand Gaussian universality (Goldt et al., 2022a; Montanari & Saeed, 2022; Dandi et al., 2023). These works suggest that simple models for the covariates can have a broad scope in the context of high-dimensional linear estimation, often mirroring real-world datasets. From a technical perspective, this phenomenon arises due to strong concentration properties in the high-dimensional regime, which imply certain universality properties of the generalization error with respect to the covariate distribution (Tao & Vu, 2010; Donoho & Tanner, 2009; Wei et al., 2022; Dudeja et al., 2023).

**Adversarial training:**    Robust empirical risk minimization, i.e. adversarial training, was first introduced for computer vision applications (Goodfellow et al., 2015a; Madry et al., 2018). A large body of work is devoted to the study of applied methods for improving its computational (Shafahi et al., 2019; Rice et al., 2020) and statistical (Zhai et al., 2019; Chen et al., 2021; Wang et al., 2023) properties. Theoretically, robust training has been considered before in both the case of Gaussian mixture models (Bhagoji et al., 2019; Dan et al., 2020; Javanmard & Soltanolkotabi, 2022) and linear regression (Raghunathan et al., 2020; Taheri et al., 2023; Dohmatob & Scetbon, 2024).

**Statistical learning theory:**    The role of regularization in statistical inference traces back to the work of Tikhonov (1963) and plays a central role in statistical learning theory, directly inspiring general inductive principles such as Structural Risk Minimization (Vapnik, 1998) and practical methods that realize this principle, such as SVMs (Cortes & Vapnik, 1995). Uniform convergence bounds, quantities that upper bound the difference between empirical and expected risk of any predictor *uniformly* inside a hypothesis class, were originally stated as a function of the VC dimension of the class (Vapnik & Chervonenkis, 1971). The Rademacher complexity of the class (Koltchinskii, 2001) is known to typically provide finer guarantees (Kakade et al., 2008). Several recent papers derive such results in the context of adversarially robust classification for linear predictors and neural networks (Yin et al., 2019; Awasthi et al., 2020; Xiao et al., 2024).

# B   Sharp High-Dimensional Asymptotics

Before delving into the technical proofs of Theorems 3.5, 3.7, 3.15 and 3.16, we provide in Table 1 a comprehensive overview of the notation used throughout the paper and particularly in these proofs. The table includes both the basic notation for the problem setup and the more specialized symbols that appear in the asymptotic analysis. We have organized the symbols thematically, starting from the fundamental quantities $(n, d, \alpha)$ and progressing to the more complex asymptotic statistics $(m^*, q^*, P^*)$. This reference should help readers track the various quantities as they appear in the detailed derivations that follow.

We now proceed with the proofs. The first theorem that will be crucial in our subsequent analysis is the Convex Gaussian MinMax Theorem (CGMT), a powerful tool in high-dimensional probability theory. The CGMT provides a connection between two seemingly unrelated optimization problems under Gaussian conditioning. Essentially, it allows us to study the properties of a complex primary optimization problem (PO) by examining a simpler auxiliary optimization problem (AO). This theorem is particularly valuable in our context as it enables us to transform intricate high-dimensional problems into more tractable lower-dimensional equivalents, significantly simplifying our analysis and leading to Theorems 3.7 and 3.16.

The CGMT states that under certain conditions, the probabilistic behavior of the primary optimization problem involving a Gaussian matrix is upper and lower bounded by the behavior of an auxiliary problem involving only Gaussian vectors. This powerful result allows us to derive tight probability bounds and asymptotic predictions for the high-dimensional estimation problems considered in this manuscript.

We state the theorem in full generality.

| Symbol | Description |
|---|---|
| $n$ | Number of training samples |
| $d$ | Input dimension |
| $\alpha = n/d$ | Sample complexity (ratio of samples to dimension) |
| $\boldsymbol{x}_i \in \mathbb{R}^d$ | Input features for sample $i$ |
| $y_i \in \{-1, +1\}$ | Binary label for sample $i$ |
| $\boldsymbol{w}_\star \in \mathbb{R}^d$ | Teacher (true) weight vector |
| $\hat{\boldsymbol{w}} \in \mathbb{R}^d$ | Learned weight vector (student) |
| $\varepsilon$ | Adversarial perturbation budget |
| $\|\cdot\|_p$ | $\ell_p$ norm, defined as $\|x\|_p = (\sum_i |x_i|^p)^{1/p}$ |
| $p^*$ | Dual exponent of $p$, satisfying $1/p + 1/p^* = 1$ |
| $E_{\mathrm{rob}}(\hat{\boldsymbol{w}})$ | Robust generalization error |
| $E_{\mathrm{gen}}(\hat{\boldsymbol{w}})$ | Standard generalization error |
| $E_{\mathrm{bnd}}(\hat{\boldsymbol{w}})$ | Boundary error (difference between robust and standard error) |
| $\lambda$ | Regularization strength parameter |
| $\widetilde{r}(w)$ | Regularization function |
| $g(\cdot)$ | Surrogate loss function |
| $\mathbb{P}_{\mathrm{out}}$ | Output channel (conditional probability of labels) |
| $P_{\mathrm{in}}$ | Input distribution |
| $P_w$ | Prior distribution on teacher weights |
| $\boldsymbol{\Sigma_\delta}$ | Matrix defining Mahalanobis norm for perturbations |
| $\boldsymbol{\Sigma_w}$ | Matrix defining Mahalanobis norm for regularization |
| $m^\star$ | Asymptotic overlap between teacher and student weights |
| $q^\star$ | Asymptotic squared $\ell_2$ norm of student weights |
| $P^\star$ | Asymptotic dual norm of student weights |

Table 1: Notation Table

**Theorem B.1** (CGMT Gordon (1988); Thrampoulidis et al. (2014)). *Let $\boldsymbol{G} \in \mathbb{R}^{m \times n}$ be an i.i.d. standard normal matrix and $\mathbf{g} \in \mathbb{R}^m$, $\mathbf{h} \in \mathbb{R}^n$ two i.i.d. standard normal vectors independent of one another. Let $\mathcal{S}_{\boldsymbol{w}}$, $\mathcal{S}_{\boldsymbol{u}}$ be two compact sets such that $\mathcal{S}_{\boldsymbol{w}} \subset \mathbb{R}^n$ and $\mathcal{S}_{\boldsymbol{u}} \subset \mathbb{R}^n$. Consider the two following optimization problems for any continuous $\psi$ on $\mathcal{S}_{\boldsymbol{w}} \times \mathcal{S}_{\boldsymbol{u}}$*

$$\mathbf{C}(\boldsymbol{G}) := \min_{\boldsymbol{w} \in \mathcal{S}_{\boldsymbol{w}}} \max_{\boldsymbol{u} \in \mathcal{S}_{\boldsymbol{u}}} \boldsymbol{u}^\top \boldsymbol{G} \boldsymbol{w} + \psi(\boldsymbol{w}, \boldsymbol{u}) \tag{21}$$

$$\mathcal{C}(\mathbf{g}, \mathbf{h}) := \min_{\boldsymbol{w} \in \mathcal{S}_{\boldsymbol{w}}} \max_{\boldsymbol{u} \in \mathcal{S}_{\boldsymbol{u}}} \|\boldsymbol{w}\|_2 \mathbf{g}^\top \boldsymbol{u} + \|\boldsymbol{u}\|_2 \mathbf{h}^\top \boldsymbol{w} + \psi(\boldsymbol{w}, \boldsymbol{u}) \tag{22}$$

*Then the following hold*

*1. For all $c \in \mathbb{R}$ we have*

$$\mathbb{P}(\mathbf{C}(\boldsymbol{G}) < c) \leq 2\mathbb{P}(\mathcal{C}(\mathbf{g}, \mathbf{h}) \leq c) \tag{23}$$

*2. Further assume that $\mathcal{S}_{\boldsymbol{w}}$ and $\mathcal{S}_{\boldsymbol{u}}$ are convex sets, $\psi$ is convex-concave on $\mathcal{S}_{\boldsymbol{w}} \times \mathcal{S}_{\boldsymbol{u}}$. Then for all $c \in \mathbb{R}$*

$$\mathbb{P}(\mathbf{C}(\boldsymbol{G}) > c) \leq 2\mathbb{P}(\mathcal{C}(\mathbf{g}, \mathbf{h}) \geq c) \tag{24}$$

*In particular for all $\mu \in \mathbb{R}$, $t > 0$ we have $\mathbb{P}(|\mathbf{C}(\boldsymbol{G}) - \mu| > t) \leq 2\mathbb{P}(|\mathcal{C}(\mathbf{g}, \mathbf{h}) - \mu| \geq t)$.*

In our analysis, we will employ a version of the CGMT applied to a general class of generalized linear models, as proved by Loureiro et al. (2021).

## B.1 Notations and Definitions

In this paper, we extensively employ the concepts of Moreau envelopes and proximal operators, pivotal elements in convex analysis frequently encountered in recent works on high-dimensional asymptotic of convex

problems (Boyd & Vandenberghe, 2004; Parikh & Boyd, 2014). For an in-depth analysis of their properties, we refer the reader to the cited literature. Here, we briefly outline their definition and the main properties for context.

**Definition B.2** (Moreau Envelope)**.** Given a convex function $f : \mathbb{R}^n \to \mathbb{R}$ we define its Moreau envelope as being

$$\mathcal{M}_{Vf(\cdot)}(\boldsymbol{\omega}) = \min_{\boldsymbol{x}} \left[ \frac{1}{2V} \|\boldsymbol{x} - \boldsymbol{\omega}\|_2^2 + f(\boldsymbol{x}) \right]. \tag{25}$$

where the Moreau envelope can be seen as a function $\mathcal{M}_{Vf(\cdot)} : \mathbb{R}^n \to \mathbb{R}$.

**Definition B.3** (Proximal Operator)**.** Given a convex function $f : \mathbb{R}^n \to \mathbb{R}$ we define its Proximal operator as being

$$\mathcal{P}_{Vf(\cdot)}(\boldsymbol{\omega}) = \arg\min_{\boldsymbol{x}} \left[ \frac{1}{2V} \|\boldsymbol{x} - \boldsymbol{\omega}\|_2^2 + f(\boldsymbol{x}) \right]. \tag{26}$$

where the Proximal operator can be seen as a function $\mathcal{P}_{Vf(\cdot)} : \mathbb{R}^n \to \mathbb{R}^n$.

**Theorem B.4** (Gradient of Moreau Envelope (Thrampoulidis et al., 2018), Lemma D1)**.** *Given a convex function $f : \mathbb{R}^n \to \mathbb{R}$, we denote its Moreau envelope by $\mathcal{M}_{Vf(\cdot)}(\cdot)$ and its Proximal operator as $\mathcal{P}_{Vf(\cdot)}(\cdot)$. Then, we have:*

$$\boldsymbol{\nabla}_{\boldsymbol{\omega}} \mathcal{M}_{Vf(\cdot)}(\boldsymbol{\omega}) = \frac{1}{V} \left( \boldsymbol{\omega} - \mathcal{P}_{Vf(\cdot)}(\boldsymbol{\omega}) \right). \tag{27}$$

Additionally we will use the following two properties

$$\mathcal{M}_{Vf(\cdot + \boldsymbol{u})}(\boldsymbol{\omega}) = \mathcal{M}_{Vf(\cdot)}(\boldsymbol{\omega} + \boldsymbol{u}), \quad \mathcal{P}_{Vf(\cdot + \boldsymbol{u})}(\boldsymbol{\omega}) = \boldsymbol{u} + \mathcal{P}_{Vf(\cdot)}(\boldsymbol{\omega} + \boldsymbol{u}), \tag{28}$$

which are easy to show from a change of variables inside the minimization.

**Definition B.5** (Dual of a Number)**.** We define the the dual of a number $a \geq 0$ as being $a^\star$ as the only number such that $1/a + 1/a^\star = 1$.

## B.2 Assumptions and Preliminary Discussion

We restate here all the assumptions that we make for the problem.

**Assumption B.6** (Estimation from the dataset)**.** Given a dataset $\mathcal{D}$ made of $n$ pairs of input outputs $\{(\boldsymbol{x}_i, y_i)\}_{i=1}^n$, where $\boldsymbol{x}_i \in \mathbb{R}^d$ and $y_i \in \mathbb{R}$ we estimate the vector $\hat{\boldsymbol{w}}$ as being

$$\hat{\boldsymbol{w}} \in \arg\min_{\boldsymbol{w} \in \mathbb{R}^d} \sum_{i=1}^n \max_{\|\boldsymbol{\delta}_i\| \leq \varepsilon} g\left( y_i \frac{\boldsymbol{w}^\top(\boldsymbol{x}_i + \boldsymbol{\delta}_i)}{\sqrt{d}} \right) + \lambda \widetilde{r}(\boldsymbol{w}), \tag{29}$$

where $g : \mathbb{R} \to \mathbb{R}$ is a convex non-increasing function, $\lambda \in [0, \infty)$ and $\widetilde{r} : \mathbb{R}^d \to \mathbb{R}$ a convex regularization function.

**Assumption B.7** (High-Dimensional Limit)**.** We consider the proportional high-dimensional regime where both the number of training data and input dimension $n, d \to \infty$ at a fixed ratio $\alpha := n/d$.

**Assumption B.8** (Regularization functions and Attack Norms considered)**.** We consider consider two settings for the perturbation norm $\|\cdot\|$ and the regularization function $r$. For the first one, the regularization function and the attack norm $\ell_p$ norms, defined as

$$\|\boldsymbol{x}\|_p = \left( \sum_{i=1}^n |x_i|^p \right)^{1/p} \tag{30}$$

for $p \in (1, \infty]$. We will refer to the index of the regularization function as $r$ and to the index of the norm inside the inner maximization as $p$ and we define $p^\star$ as the dual number of $p$ (definition B.5).

For the second case, both the regularization function and the attack norm are Mahalanobis norms, defined as

$$\|\boldsymbol{x}\|_{\boldsymbol{\Sigma}} = \sqrt{\boldsymbol{x}^\top \boldsymbol{\Sigma} \boldsymbol{x}} \tag{31}$$

for a positive definite matrix $\mathbf{\Sigma}$. We refer to the index of the matrix of the regularization function as $\mathbf{\Sigma_w}$ and to the matrix of the norm inside the inner maximization as $\mathbf{\Sigma_\delta}$. In this case, we define $p = r = 2$ (in order to unify notations) and we will thus talk about $p^\star = r^\star = 2$.

This setting considers most of the losses used in machine learning setups for binary classification, *e.g.* logistic, hinge, exponential losses. We additionally remark that with the given choice of regularization the whole cost function is coercive.

**Assumption B.9** (Scaling of Adversarial Norm Constraint). We suppose that the value of $\varepsilon$ scales with the dimension $d$ such that $\varepsilon/(\sqrt{d} \sqrt[p^\star]{d}) = O_d(1)$.

**Assumption B.10** (Data Distribution). We consider two cases of data distribution. Both of them will rely on the following general generative process. For each $i \in [n]$, the covariates $\boldsymbol{x}_i \in \mathbb{R}^d$ are drawn i.i.d. from a data distribution $P_{\text{in}}(\boldsymbol{x})$. Then, the corresponding $y_i$ is sampled independently from the conditional distribution $P_{\text{out}}$. More succinctly, one can write the data distribution for a given pair $(\boldsymbol{x}, y)$ as

$$P(\boldsymbol{x}, y) = \int_{\mathbb{R}^d} \mathrm{d}\boldsymbol{w}_\star \mathbb{P}_{\text{out}} \left( y \left| \frac{\langle \boldsymbol{w}_\star, \boldsymbol{x} \rangle}{\sqrt{d}} \right. \right) P_{\text{in}}(\boldsymbol{x}) P_{\boldsymbol{w}}(\boldsymbol{w}_\star), \tag{32}$$

The target weight vector $\boldsymbol{w}_\star \in \mathbb{R}^d$ is drawn from a prior probability distribution $P_{\boldsymbol{w}}$.

Our two cases differentiate in the following way. For the first case, we consider $P_{\text{in}}(\boldsymbol{x}) = \mathcal{N}_{\boldsymbol{x}}(\mathbf{0}, \text{Id}_d)$ and $P_{\boldsymbol{w}}$ which is separable, i.e. $P_{\boldsymbol{w}}(\boldsymbol{w}) = \prod_{i=1}^d P_w(w_i)$ for a distribution $P_w$ in $\mathbb{R}$ with finite variance $\text{Var}(P_w) = \rho < \infty$.

For the second case, we consider $P_{\text{in}}(\boldsymbol{x}) = \mathcal{N}_{\boldsymbol{x}}(\mathbf{0}, \mathbf{\Sigma_x})$ and $P_{\boldsymbol{w}}(\boldsymbol{w}) = \mathcal{N}_{\boldsymbol{w}}(\mathbf{0}, \mathbf{\Sigma_\theta})$.

**Assumption B.11** (Limiting Convergence of Spectral Values). We suppose that $\mathbf{\Sigma_x}, \mathbf{\Sigma_\delta}, \mathbf{\Sigma_\theta}, \mathbf{\Sigma_w}$ are simultaneously diagonalisable. We call $\mathbf{\Sigma_x} = \mathrm{S}^\top \text{diag}(\omega_i) \mathrm{S}$, $\mathbf{\Sigma_\delta} = \mathrm{S}^\top \text{diag}(\zeta_i) \mathrm{S}$ and $\mathbf{\Sigma_w} = \mathrm{S}^\top \text{diag}(w_i) \mathrm{S}$. We define $\bar{\boldsymbol{\theta}} = \mathrm{S} \mathbf{\Sigma_x}^\top \boldsymbol{w}_\star / \sqrt{\rho}$. We assume that the empirical distributions of eigenvalues and the entries of $\bar{\boldsymbol{\theta}}$ jointly converge to a probability distribution $\mu$ as

$$\sum_{i=1}^d \delta(\bar{\boldsymbol{\theta}}_i - \bar{\theta}) \delta(\omega_i - \omega) \delta(\zeta_i - \zeta) \delta(w_i - w) \to \mu. \tag{33}$$

## B.3 Problem Simplification

Recall that we start from the following optimization problem:

$$\Phi_d = \min_{\boldsymbol{w} \in \mathbb{R}^d} \sum_{i=1}^n \max_{\|\boldsymbol{\delta}_i\| \leq \varepsilon} g \left( y_i \frac{\boldsymbol{w}^\top (\boldsymbol{x}_i + \boldsymbol{\delta}_i)}{\sqrt{d}} \right) + \lambda \widetilde{r}(\boldsymbol{w}), \tag{34}$$

where $\widetilde{r}(\cdot)$ is a convex regularization function and $g(\cdot)$ is a non-increasing loss function. The non-increasing property of $g$ allows us to simplify the inner maximization, leading to an equivalent formulation

$$\Phi_d = \min_{\boldsymbol{w} \in \mathbb{R}^d} \sum_{i=1}^n g \left( y_i \frac{\boldsymbol{w}^\top \boldsymbol{x}_i}{\sqrt{d}} - \frac{\varepsilon}{\sqrt{d}} \|\boldsymbol{w}\|_\star \right) + \lambda \widetilde{r}(\boldsymbol{w}). \tag{35}$$

To facilitate our analysis, we introduce auxiliary variables $P = \|\boldsymbol{w}\|_\star^{p^\star} / d$ and $\hat{P}$ (the Lagrange parameter relative to this variable), which allow us to decouple the norm constraints. This leads to a min-max formulation

$$\Phi_d = \min_{\boldsymbol{w} \in \mathbb{R}^d, P} \max_{\hat{P}} \sum_{i=1}^n g \left( y_i \frac{\boldsymbol{w}^\top \boldsymbol{x}_i}{\sqrt{d}} - \frac{\varepsilon}{\sqrt[p^\star]{d}} \sqrt[p^\star]{P} \right) + \lambda \widetilde{r}(\boldsymbol{w}) + \hat{P} \|\boldsymbol{w}\|_\star^{p^\star} - dP\hat{P}, \tag{36}$$

where we switched the value of $\varepsilon$ for its value without the scaling in $d$. This reformulation is what will allow us to apply the CGMT in subsequent steps.

It's worth noting the significance of the scaling for $\varepsilon$ as detailed in Assumption B.9. In the high-dimensional limit $d \to \infty$, it's essential that all terms in $\Phi_d$ exhibit the same scaling with respect to $d$. This careful scaling ensures that our asymptotic analysis remains well-behaved and meaningful in the high-dimensional regime.

### B.4 Scalarization and Application of CGMT

To facilitate our analysis, we further introduce effective regularization and loss functions, $\widetilde{\widetilde{r}}$ and $\widetilde{g}$, respectively. These functions are defined as

$$\widetilde{g}(\boldsymbol{y}, \boldsymbol{z}) = \sum_{i=1}^{n} g\left( y_i z_i - \frac{\varepsilon}{\sqrt[p^\star]{d}} \sqrt[p^\star]{P} \right), \quad \widetilde{\widetilde{r}}(\boldsymbol{w}) = \widetilde{r}(\boldsymbol{w}) + \hat{P}\|\boldsymbol{w}\|_{p^\star}^{p^\star}. \tag{37}$$

A crucial step in our analysis involves inverting the order of the min-max optimization. We can justify this operation by considering the minimization with respect to $\boldsymbol{w} \in \mathbb{R}^d$ at fixed values of $\hat{P}$ and $P$. This reordering is valid due to the convexity of our original problem. Specifically, the objective function is convex in $\boldsymbol{w}$ and concave in $\hat{P}$ and $P$, and the constraint sets are convex. Under these conditions, we apply Sion's minimax theorem, which guarantees the existence of a saddle point and allows us to interchange the order of minimization and maximization without affecting the optimal value.

This reformulation enables us to directly apply (Loureiro et al., 2021, Lemma 11). This lemma represents a meticulous application of Theorem B.1 to scenarios involving non-separable convex regularization and loss functions. The result is a lower-dimensional equivalent of our original high-dimensional minimization problem that represent the limiting behavior of the solution of the high-dimensional problem.

Consequently, our analysis now focuses on a low-dimensional functional, which takes the form

$$\widetilde{\Phi} = \min_{P, m, \eta, \tau_1} \max_{\hat{P}, \kappa, \tau_2, \nu} \left[ \frac{\kappa \tau_1}{2} - \alpha \mathcal{L}_g - \frac{\eta}{2\tau_2}\left( \nu^2 \rho + \kappa^2 \right) - \frac{\eta \tau_2}{2} - \mathcal{L}_{\widetilde{r}} + m\nu - P\hat{P} \right] \tag{38}$$

where we have restored the min max order of the problem.

In this expression, $\boldsymbol{g}$ and $\boldsymbol{h}$ are independent Gaussian vectors with i.i.d. standard normal components. The terms $\mathcal{L}_g$ and $\mathcal{L}_{\widetilde{r}}$ represent the scaled averages of Moreau Envelopes (eq. (25))

$$\mathcal{L}_g = \frac{1}{n} \mathbb{E}\left[ \mathcal{M}_{\frac{\tau_1}{\kappa}\widetilde{g}(\boldsymbol{y}, \cdot)} \left( \frac{m}{\sqrt{\rho}}\boldsymbol{s} + \eta \boldsymbol{h} \right) \right] \tag{39}$$

$$\mathcal{L}_{\widetilde{r}} = \frac{1}{d} \mathbb{E}\left[ \mathcal{M}_{\frac{\eta}{\tau_2}\widetilde{\widetilde{r}}(\cdot)} \left( \frac{\eta}{\tau_2}(\kappa \boldsymbol{g} + \nu \boldsymbol{w}_\star) \right) \right] \tag{40}$$

The extremization problem in eq. (38) is related to the original optimization problem in eq. (34) as it can be thought as the leading part in the limit $n, d \to \infty$.

This dimensional reduction is the step that allows us to study the asymptotic properties of our original high-dimensional problem through a more tractable low-dimensional optimization and thus have in the end a low dimensional set of equations to study.

It's important to note that the optimization problem $\widetilde{\Phi}$ is still implicitly defined in terms of the dimension $d$ and, consequently, as a function of the sample size $n$. We introduce two variables

$$\boldsymbol{w}_{\text{eq}} = \mathcal{P}_{\frac{\eta^*}{\tau_2^*}\widetilde{\widetilde{r}}(\cdot)} \left( \frac{\eta^*}{\tau_2^*}\left( \nu^* \mathbf{t} + \kappa^* \mathbf{g} \right) \right), \quad \boldsymbol{z}_{\text{eq}} = \mathcal{P}_{\frac{\tau_1^*}{\kappa^*}\widetilde{g}(\cdot, \mathbf{y})} \left( \frac{m^*}{\sqrt{\rho}}\mathbf{s} + \eta^* \mathbf{h} \right) \tag{41}$$

where $(\eta^\star, \tau_2^\star, P^\star, \hat{P}^\star, \kappa^\star, \nu^\star, m^\star, \tau_1^\star)$ are the extremizer points of $\widetilde{\Phi}$.

Building upon (Loureiro et al., 2021, Theorem 5), we can establish a convergence result. Let $\hat{\boldsymbol{w}}$ be an optimal solution of the problem defined in eq. (34), and let $\hat{\boldsymbol{z}} = \frac{1}{\sqrt{d}}\boldsymbol{X}\hat{\boldsymbol{w}}$. For any Lipschitz function $\varphi_1 : \mathbb{R}^d \to \mathbb{R}$, and any separable, pseudo-Lipschitz function $\varphi_2 : \mathbb{R}^n \to \mathbb{R}$, there exist constants $\epsilon, C, c > 0$ such that

$$\mathbb{P}\left( \left| \phi_1\left( \frac{\hat{\boldsymbol{w}}}{\sqrt{d}} \right) - \mathbb{E}\left[ \phi_1\left( \frac{\boldsymbol{w}_{\text{eq}}}{\sqrt{d}} \right) \right] \right| \geq \epsilon \right) \leq \frac{C}{\epsilon^2} e^{-cn\epsilon^4}$$

$$\mathbb{P}\left( \left| \phi_2\left( \frac{\hat{\boldsymbol{z}}}{\sqrt{n}} \right) - \mathbb{E}\left[ \phi_2\left( \frac{\boldsymbol{z}_{\text{eq}}}{\sqrt{n}} \right) \right] \right| \geq \epsilon \right) \leq \frac{C}{\epsilon^2} e^{-cn\epsilon^4} \tag{42}$$

It demonstrates that the limiting values of any function depending on $\hat{w}$ and $\hat{z}$ can be computed by taking the expectation of the same function evaluated at $w_{\text{eq}}$ or $z_{\text{eq}}$, respectively. This convergence property allows us to translate results from our low-dimensional proxy problem back to the original high-dimensional setting with high probability.

## B.5 Derivation of Saddle Point equations

We now want to show that extremizing the values of $m, \eta, \tau_1, P, \hat{P}, \nu, \tau_2, \kappa$ lead to the optimal value $\widetilde{\Phi}$ of eq. (38). We are going to directly derive the saddle point equations and then argue that in the high-dimensional limit they become exactly the ones reported in the main text.

We obtain the first set of derivatives that depend only on the loss function and the channel part by taking the derivatives with respect to $m, \eta, \tau_1, P$ to obatin

$$
\begin{aligned}
\frac{\partial}{\partial m} &: \nu = \alpha \frac{\kappa}{n\tau_1} \mathbb{E}\left[\left(\frac{m}{\eta\rho}\mathbf{h} - \frac{\mathbf{s}}{\sqrt{\rho}}\right)^\top \mathcal{P}_{\frac{\tau_1}{\kappa}\widetilde{g}(\cdot,\mathbf{y})}\left(\frac{m}{\sqrt{\rho}}\mathbf{s} + \eta\mathbf{h}\right)\right] \\
\frac{\partial}{\partial \eta} &: \tau_2 = \alpha\frac{\kappa}{\tau_1}\eta - \frac{\kappa\alpha}{\tau_1 n}\mathbb{E}\left[\mathbf{h}^\top \mathcal{P}_{\frac{\tau_1}{\kappa}\widetilde{g}(\cdot,\mathbf{y})}\left(\frac{m}{\sqrt{\rho}}\mathbf{s} + \eta\mathbf{h}\right)\right] \\
\frac{\partial}{\partial \tau_1} &: \frac{\tau_1^2}{2} = \frac{1}{2}\alpha\frac{1}{n}\mathbb{E}\left[\left\|\frac{m}{\sqrt{\rho}}\mathbf{s} + \eta\mathbf{h} - \mathcal{P}_{\frac{\tau_1}{\kappa}\widetilde{g}(\cdot,y)}\left(\frac{m}{\sqrt{\rho}}\mathbf{s} + \eta\mathbf{h}\right)\right\|_2^2\right] \\
\frac{\partial}{\partial P} &: \hat{P} = \frac{\alpha}{n}\partial_P\mathbb{E}\left[\mathcal{M}_{\frac{\tau_1}{\kappa}\widetilde{g}(\mathbf{y},\cdot)}\left(\frac{m}{\sqrt{\rho}}\mathbf{s} + \eta\mathbf{h}\right)\right]
\end{aligned}
\tag{43}
$$

By taking the derivatives with respect to the remaining variables $\kappa, \nu, \tau_2, \hat{P}$ we obtain a set of equations depending on regularization and prior over the teacher weights

$$
\begin{aligned}
\frac{\partial}{\partial \kappa} &: \tau_1 = \frac{1}{d}\mathbb{E}\left[\mathbf{g}^\top \mathcal{P}_{\frac{\eta}{\tau_2}\widetilde{r}(\cdot)}\left(\frac{\eta}{\tau_2}\left(\nu w_\star + \kappa\mathbf{g}\right)\right)\right] \\
\frac{\partial}{\partial \nu} &: m = \frac{1}{d}\mathbb{E}\left[w_\star^\top \mathcal{P}_{\frac{\eta}{\tau_2}\widetilde{r}(\cdot)}\left(\frac{\eta}{\tau_2}\left(\nu w_\star + \kappa\mathbf{g}\right)\right)\right] \\
\frac{\partial}{\partial \tau_2} &: \frac{1}{2d}\frac{\tau_2}{\eta}\mathbb{E}\left[\left\|\frac{\eta}{\tau_2}\left(\nu w_\star + \kappa\mathbf{g}\right) - \mathcal{P}_{\frac{\eta}{\tau_2}\widetilde{r}(\cdot)}\left(\frac{\eta}{\tau_2}\left(\nu w_\star + \kappa\mathbf{g}\right)\right)\right\|_2^2\right] = \frac{\eta}{2\tau_2}\left(\nu^2\rho + \kappa^2\right) - m\nu - \kappa\tau_1 + \frac{\eta\tau_2}{2} + \frac{\tau_2}{2\eta}\frac{m^2}{\rho} \\
\frac{\partial}{\partial \hat{P}} &: P = \frac{1}{d}\partial_{\hat{P}}\mathbb{E}\left[\mathcal{M}_{\frac{\eta}{\tau_2}\widetilde{r}(\cdot)}\left(\frac{\eta}{\tau_2}\left(\kappa\mathbf{g} + \nu w_\star\right)\right)\right]
\end{aligned}
\tag{44}
$$

The rewriting of these equations in the desired form in Theorems 3.7 and 3.16 follows from the same considerations as in (Loureiro et al., 2021, Appendix C.2).

To perform this rewriting the first ingredient we need is the following change of variables

$$
\begin{aligned}
m &\leftarrow m, & q &\leftarrow \eta^2 + \frac{m^2}{\rho}, & V &\leftarrow \frac{\tau_1}{\kappa}, & P &\leftarrow P, \\
\hat{V} &\leftarrow \frac{\tau_2}{\eta}, & \hat{q} &\leftarrow \kappa^2, & \hat{m} &\leftarrow \nu, & \hat{P} &\leftarrow \hat{P}.
\end{aligned}
\tag{45}
$$

ant the use of Isserlis' theorem (Isserlis, 1918) to simplify the expectation where Gaussian $g, h$ vectors are present.

### B.5.1 Rewriting of the Channel Saddle Points

To obtain specifically the form implied in the main text we introduce

$$
\mathcal{Z}_0(y, \omega, V) = \int \frac{dx}{\sqrt{2\pi V}} e^{-\frac{1}{2V}(x-\omega)^2} \delta\left(y - f^0(x)\right),
\tag{46}
$$

where this definition is equivalent to the one presented in eq. (10). The function $\mathcal{Z}_0$ can be interpreted as a partition function of the conditional distribution $\mathbb{P}_{\text{out}}$ and contains all of the information about the label generating process.

### B.5.2 Specialization of Prior Saddle Points for $\ell_p$ norms

In the case of $\ell_p$ norms, we can leverage the separable nature of the regularization to simplify our equations. The key insight here is that the proximal operator of a separable regularization is itself separable. This property allows us to treat each dimension independently, leading to a significant simplification of our high-dimensional problem.

First, due to the separability, all terms depending on the proximal of either $\widetilde{g}$ or $\widetilde{\widetilde{r}}$ simplify the $n$ or $d$ at the denominator. This cancellation is crucial as it eliminates the explicit dependence on the problem dimension, allowing us to derive dimension-independent equations.

Next, we introduce

$$\mathcal{Z}_{\text{w}}(\gamma, \Lambda) = \int \mathrm{d}w\, P_{\text{w}}(w) e^{-\frac{\Lambda}{2}w^2 + \gamma w}, \tag{47}$$

which, in turn, leads in the form shown in eq. (9).

### B.5.3 Specialization of Prior Saddle Points for Mahalanobis norms

In the case of Mahalanobis norm, the form of the proximal of the effective regularization function is specifically

$$\mathcal{P}_{V\widetilde{r}(\cdot)}(\boldsymbol{\omega}) = \arg\min_{\boldsymbol{z}} \left[ \lambda \boldsymbol{z}^\top \boldsymbol{\Sigma_w} \boldsymbol{z} + \hat{P} \boldsymbol{z}^\top \boldsymbol{\Sigma_\delta} \boldsymbol{z} + \frac{1}{2V} \|\boldsymbol{z} - \boldsymbol{\omega}\|_2^2 \right] = \frac{1}{V}\left( 2\hat{P}\boldsymbol{\Sigma_\delta} + 2\lambda\boldsymbol{\Sigma_w} + \frac{1}{V} \right)^{-1} \boldsymbol{\omega} \tag{48}$$

By substituting this explicit form into the equations from eq. (44), we obtain a set of simplified equations that still depends on the dimension

$$
\begin{aligned}
m &= \frac{1}{d}\operatorname{tr}\left[\hat{m}\boldsymbol{\Sigma_x^\top}\boldsymbol{\theta_0}\boldsymbol{\theta_0^\top}\boldsymbol{\Sigma_x}\left(\lambda\boldsymbol{\Sigma_w} + \hat{P}\boldsymbol{\Sigma_\delta} + \hat{V}\boldsymbol{\Sigma_x}\right)^{-1}\right] \\
q &= \frac{1}{d}\operatorname{tr}\left[\left(\hat{m}^2\boldsymbol{\Sigma_x^\top}\boldsymbol{\theta_0}\boldsymbol{\theta_0^\top}\boldsymbol{\Sigma_x} + \hat{q}\boldsymbol{\Sigma_x}\right)\boldsymbol{\Sigma_x}\left(\lambda\boldsymbol{\Sigma_w} + \hat{P}\boldsymbol{\Sigma_\delta} + \hat{V}\boldsymbol{\Sigma_x}\right)^{-2}\right] \\
V &= \frac{1}{d}\operatorname{tr}\left[\boldsymbol{\Sigma_x}\left(\lambda\boldsymbol{\Sigma_w} + \hat{P}\boldsymbol{\Sigma_\delta} + \hat{V}\boldsymbol{\Sigma_x}\right)^{-1}\right] \\
P &= \frac{1}{d}\operatorname{tr}\left[\left(\hat{m}^2\boldsymbol{\Sigma_x^\top}\boldsymbol{\theta_0}\boldsymbol{\theta_0^\top}\boldsymbol{\Sigma_x} + \hat{q}\boldsymbol{\Sigma_x}\right)\boldsymbol{\Sigma_\delta}\left(\lambda\boldsymbol{\Sigma_w} + \hat{P}\boldsymbol{\Sigma_\delta} + \hat{V}\boldsymbol{\Sigma_x}\right)^{-2}\right]
\end{aligned}
\tag{49}
$$

The final step involves taking the high-dimensional limit of these equations. Here, we leverage our assumptions about the trace of the relevant matrices to further simplify the expressions so that they only depend on the limiting distribution $\mu$ from Assumption B.11.

Specifically, the assumptions on the trace allow us to replace certain high-dimensional operations with scalar quantities, effectively reducing the dimensionality of our problem. This dimensionality reduction is crucial for obtaining tractable equations in the high-dimensional limit. In the end we obtain the equations in eq. (14).

### B.6 Different channels and Prior functions

We want to show how the different functions $\mathcal{Z}_0, \mathcal{Z}_{\text{w}}$ look like for some choices of output channel and prior in the data model. For the case of a probit output channel, we have by direct calculation

$$\mathcal{Z}_0(y, \omega, V) = \frac{1}{2}\operatorname{erfc}\left(-y\frac{\omega}{\sqrt{2(V + \tau^2)}}\right) \tag{50}$$

For the case of a channel of the form $y = \text{sign}(z) + \sqrt{\Delta^*}\xi$, one has that

$$\mathcal{Z}_0(y,\omega,V) = \mathcal{N}_y(1,\Delta^\star)\frac{1}{2}\left(1 + \text{erf}\left(\frac{\omega}{\sqrt{2V}}\right)\right) + \mathcal{N}_y(-1,\Delta^\star)\frac{1}{2}\left(1 - \text{erf}\left(\frac{\omega}{\sqrt{2V}}\right)\right) \tag{51}$$

For the choices of the prior over the teacher weights, we have for a Gaussian prior that

$$\mathcal{Z}_{\text{w}}(\gamma,\Lambda) = \frac{1}{\sqrt{\Lambda+1}}e^{\gamma^2/2(\Lambda+1)} \tag{52}$$

or for sparse binary weights

$$\mathcal{Z}_{\text{w}}(\gamma,\Lambda) = \rho + e^{-\frac{\Lambda}{2}}(1-\rho)\cosh(\gamma) \tag{53}$$

### B.7 Error Metrics

To derive the form of the generalization error the procedure is the same as detailed in Aubin et al. (2020) or in Mignacco et al. (2020, Appendix A). We report here the final form being

$$E_{\text{gen}} = \frac{1}{\pi}\arccos\left(\frac{m}{\sqrt{(\rho+\tau^2)q}}\right) \tag{54}$$

To derive the form for the boundary error one can proceed in the same way as (Gerace et al., 2021, Appendix D) and obtain

$$E_{\text{bnd}} = \int_0^{\varepsilon^{\,p^\star\sqrt{P}/\sqrt{q}}} \text{erfc}\left(-\frac{m}{\sqrt{q}}\lambda\frac{1}{\sqrt{2(\rho+\tau^2-\frac{m^2}{q})}}\right)\frac{e^{-\frac{1}{2}\lambda^2}}{\sqrt{2\pi}}\,\mathrm{d}\lambda \tag{55}$$

We are also interested in the *average teacher margin* defined as

$$\mathbb{E}\left[y\boldsymbol{w}_\star^\top\boldsymbol{x}\right] \tag{56}$$

which can be expressed as a function of the solutions of the saddle point equations as follows:

$$\sqrt{\frac{2}{\pi}}\frac{\sqrt{\rho}}{\sqrt{1+\frac{\tau^2}{\rho}}} \tag{57}$$

### B.8 Asymptotic in the low sample complexity regime

This section examines the asymptotic behavior of our model in the regime of low sample complexity. Our analysis is motivated by numerical observations of the overlaps $m, q, P, V$ in the small $\alpha$ regime, as illustrated in Figure 3.

Based on these observations, we propose a general scaling ansatz for the overlap parameters (solutions of the equations presented in Theorems 3.7 and 3.16) as functions of the sample complexity $\alpha$

$$m^\star = m_0\alpha^{\delta_m}, \quad q^\star = q_0\alpha^{\delta_q}, \quad V^\star = V_0\alpha^{\delta_V}, \quad P^\star = P_0\alpha^{\delta_P}, \tag{58}$$

where the values with a zero subscript do not depend on $\alpha$ and the exponents are all positive. We focus on the noiseless case $\tau = 0$.

We are interested in the expansion of the generalization error and the boundary error, keeping only the most relevant terms in the limit $\alpha \to 0^+$. For the generalization error we have

$$E_{\text{gen}} = \frac{1}{\pi}\arccos\left(\frac{m^\star}{\sqrt{\rho q^\star}}\right) = \frac{1}{2} - \frac{m_0}{\pi\sqrt{\rho q_0}}\alpha^{\delta_m - \frac{\delta_q}{2}} + o\left(\alpha^{\delta_m - \delta_q/2}\right) \tag{59}$$

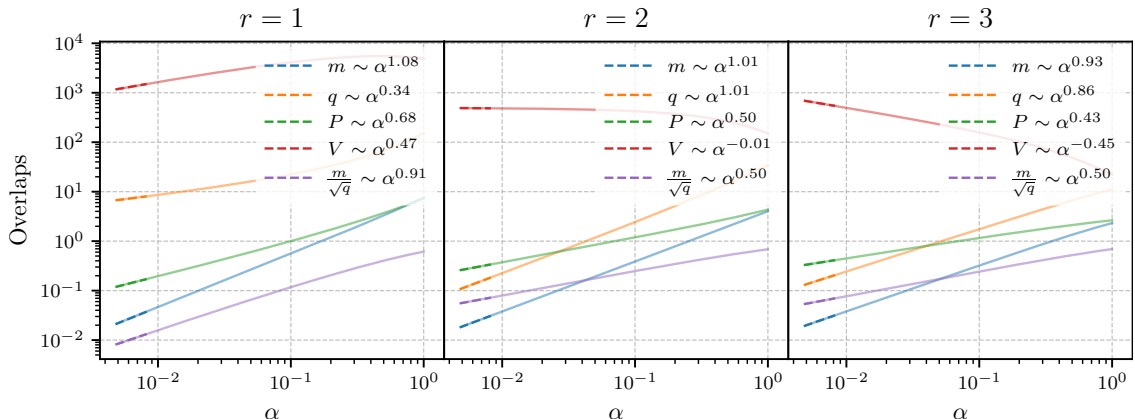

Figure 3: Scaling of the overlap parameters in the low sample complexity regime for $p = \infty$, $\varepsilon = 0.3$, $\rho = 1$ and $\lambda = 10^{-3}$. The numbers presented in the legends are the linear fit in log-log scale of the dashed part.

and for the boundary error a similar expansion leads to

$$E_{\text{bnd}} = \int_0^{\frac{p^\star\sqrt{P^\star}}{q^\star}} \text{erfc}\left(\frac{-\frac{m^\star}{\sqrt{q^\star}}\nu}{\sqrt{2(\rho - (m^\star)^2/q^\star)}}\right)\frac{e^{-\frac{\nu^2}{2}}}{\sqrt{2\pi}}\,\text{d}\nu \tag{60}$$

$$= \frac{\varepsilon_g\,^{p^\star}\sqrt{P_0}}{\sqrt{2\pi q_0}}\alpha^{\delta_P/p^\star - \delta_q/2} + \frac{\theta_0}{2\pi}\alpha^{2\delta_P/p^\star + \delta_m - 2\delta_q} + o(\alpha^\kappa) \tag{61}$$

where $\kappa = \max(\delta_P/p^\star - \delta_q/2, 2\delta_P/p^\star + \delta_m - 2\delta_q)$.

Numerical simulations reveal a clear distinction in the low $\alpha$ regime between cases where the regularization parameter $r = p^\star$ and $r \neq p^\star$. Figure 3 illustrates this difference for a fixed regularization parameter $\lambda$. We identify two scenarios that characterize the behavior of the leading term in the boundary error expansion

**When $\delta_P/p^\star > \delta_q/2$:** This occurs when $p^\star = r = 1$. In this case, the leading term has a positive exponent, causing it to vanish as $\alpha \to 0$.

**When $\delta_P/p^\star = \delta_q/2$:** This scenario arises when $r \neq p^\star = 1$. Here, the exponent of the leading term becomes zero, resulting in a constant term independent of $\alpha$.

Notably, in all cases we've examined, the second terms in both the generalization error and boundary error expansions consistently approach zero in the limit of low sample complexity.

## C    Rademacher Complexity Analysis

This appendix contains detailed proofs of the remaining results presented in the paper. For the reader's convenience, we restate each statement before its proof.

**Proposition C.1.** *Let $\varepsilon, \sigma > 0$. Consider a sample $\mathcal{S} = \{(\boldsymbol{x}_1, y_1), \ldots, (\boldsymbol{x}_n, y_n)\}$, and let $\mathcal{H}_{\widetilde{r}}$ be the hypothesis class defined in eq. (15). Then, it holds:*

$$\hat{\mathfrak{R}}_{\text{S}}(\widetilde{\mathcal{H}_{\widetilde{r}}}) \leq \max_{i \in [n]}\,_r\|\mathbf{x}_i\|_\star \mathcal{W}_{\widetilde{r}}\sqrt{\frac{2}{\sigma n}} + \frac{\varepsilon}{2\sqrt{n}}\sup_{\boldsymbol{w}:\widetilde{r}(\boldsymbol{w})\leq\mathcal{W}_{\widetilde{r}}^2}\|\boldsymbol{w}\|_\star, \tag{62}$$

*where $_r\|\cdot\|_\star, \|\cdot\|_\star$ denote the dual norm of $_r\|\cdot\|, \|\cdot\|$, respectively.*

*Proof.* We have:

$$
\begin{aligned}
\hat{\Re}_{\mathrm{S}}(\widetilde{\mathcal{H}_{\widetilde{r}}}) &= \mathbb{E}_\sigma \left[ \frac{1}{n} \sup_{h \in \mathcal{H}_{\widetilde{r}}} \sum_{i=1}^n \sigma_i \min_{\|\mathbf{x}_i' - \mathbf{x}_i\| \le \varepsilon} y_i h(\mathbf{x}_i') \right] \\
&= \mathbb{E}_\sigma \left[ \frac{1}{n} \sup_{h \in \mathcal{H}_{\widetilde{r}}} \sum_{i=1}^n \sigma_i y_i \langle \boldsymbol{w}, \mathbf{x}_i \rangle - \varepsilon \|\boldsymbol{w}\|_\star \right] && \text{(Def. of dual norm)} \\
&\le \mathbb{E}_\sigma \left[ \frac{1}{n} \sup_{h \in \mathcal{H}_{\widetilde{r}}} \sum_{i=1}^n \sigma_i y_i \langle \boldsymbol{w}, \mathbf{x}_i \rangle \right] + \mathbb{E}_\sigma \left[ \frac{1}{n} \sup_{h \in \mathcal{H}_{\widetilde{r}}} \sum_{i=1}^n -\varepsilon \sigma_i \|\boldsymbol{w}\|_\star \right] && \text{(Subadditivity of supremum)} \\
&= \hat{\Re}_{\mathrm{S}}(\mathcal{H}_{\widetilde{r}}) + \frac{\varepsilon}{2} \mathbb{E}_\sigma \left[ \left| \frac{1}{n} \sum_{i=1}^n \sigma_i \right| \right] \sup_{\boldsymbol{w}: \widetilde{r}(\boldsymbol{w}) \le \mathcal{W}_r^2} \|\boldsymbol{w}\|_\star . && \text{(Symmetry of } \sigma)
\end{aligned}
\tag{63}
$$

For the first term, i.e. the "clean" Rademacher Complexity, we plug in (Kakade et al., 2008, Theorem 1). By Jensen's inequality, we have for the second term:

$$
\mathbb{E}_\sigma \left[ \left| \frac{1}{n} \sum_{i=1}^n \sigma_i \right| \right] \le \sqrt{ \mathbb{E}_\sigma \left[ \left( \frac{1}{n} \sum_{i=1}^n \sigma_i \right)^2 \right] } = \frac{1}{\sqrt{n}},
\tag{64}
$$

which concludes the proof. $\qquad \square$

**Corollary C.2.** *Let $\varepsilon > 0$. Then:*

$$
\hat{\Re}_{\mathrm{S}}(\widetilde{\mathcal{H}}_{\|\cdot\|_2^2}) \le \frac{\max_{i \in [n]} \|\mathbf{x}_i\|_2 \mathcal{W}_2}{\sqrt{n}} + \frac{\varepsilon \mathcal{W}_2}{2\sqrt{n}} \sqrt{\lambda_{\min}^{-1}(\boldsymbol{\Sigma}_{\boldsymbol{\delta}})}.
\tag{65}
$$

*Proof.* Leveraging Proposition 4.2, the first term of the RHS follows from the fact that the squared $\ell_2$ norm is 1-strongly convex (w.r.t itself). For the second term, we have that the dual norm of $\|\cdot\|_{\boldsymbol{\Sigma}_{\boldsymbol{\delta}}}$ is given by $\|\cdot\|_{\boldsymbol{\Sigma}_{\boldsymbol{\delta}}^{-1}} = \sqrt{\langle \boldsymbol{w}, \boldsymbol{\Sigma}_{\boldsymbol{\delta}}^{-1} \boldsymbol{w} \rangle}$. Then, it holds:

$$
\begin{aligned}
\sup_{\boldsymbol{w}: \|\boldsymbol{w}\|_2^2 \le \mathcal{W}_2^2} \|\boldsymbol{w}\|_\star &= \sup_{\boldsymbol{w}: \|\boldsymbol{w}\|_2^2 \le \mathcal{W}_2^2} \|\boldsymbol{w}\|_{\boldsymbol{\Sigma}_{\boldsymbol{\delta}}^{-1}} \\
&= \mathcal{W}_2 \sup_{\boldsymbol{w}: \|\boldsymbol{w}\|_2 \le 1} \|\boldsymbol{w}\|_{\boldsymbol{\Sigma}_{\boldsymbol{\delta}}^{-1}} \\
&= \mathcal{W}_2 \sqrt{\lambda_{\max}(\boldsymbol{\Sigma}_{\boldsymbol{\delta}}^{-1})},
\end{aligned}
\tag{66}
$$

where the last equality follows from Courant–Fischer–Weyl's min-max principle. $\qquad \square$

**Corollary C.3.** *Let $\boldsymbol{\Sigma}_{\boldsymbol{w}} = \sum_{i=1}^d \alpha_i \mathbf{v}_i \mathbf{v}_i^T$ and $\boldsymbol{\Sigma}_{\boldsymbol{\delta}} = \sum_{i=1}^d \lambda_i \mathbf{v}_i \mathbf{v}_i^T$, with $\mathbf{v}_i \in \mathbb{R}^d$ being orthonormal. Then:*

$$
\hat{\Re}_{\mathrm{S}}(\widetilde{\mathcal{H}}_{\|\cdot\|_A^2}) \le \frac{\mathcal{W}_A \max_{i \in [n]} \|\mathbf{x}_i\|_{\boldsymbol{\Sigma}_{\boldsymbol{w}}^{-1}}}{\sqrt{n}} + \frac{\varepsilon \mathcal{W}_A}{2\sqrt{n}} \sqrt{\max_{i \in [d]} \frac{1}{\lambda_i \alpha_i}}.
\tag{67}
$$

*Proof.* For the worst-case part, we have:

$$\sup_{\boldsymbol{w}:\|\boldsymbol{w}\|_{\boldsymbol{\Sigma_w}}^2 \leq \mathcal{W}_A^2} \|\boldsymbol{w}\|_{\Sigma^{-1}} = \mathcal{W}_A \sup_{\boldsymbol{w}:\|\boldsymbol{w}\|_{\boldsymbol{\Sigma_w}} \leq 1} \sqrt{\langle \boldsymbol{w}, \boldsymbol{\Sigma_\delta}^{-1} \boldsymbol{w} \rangle}$$

$$= \mathcal{W}_A \sup_{\boldsymbol{w}:\|\boldsymbol{w}\|_{\boldsymbol{\Sigma_w}} \leq 1} \sqrt{\sum_{i=1}^d \lambda_i^{-1} \langle \boldsymbol{w}, \mathbf{v}_i \rangle^2}$$

$$= \mathcal{W}_A \sup_{\boldsymbol{w}:\|\boldsymbol{w}\|_{\boldsymbol{\Sigma_w}} \leq 1} \sqrt{\sum_{i=1}^d \frac{\lambda_i^{-1}}{\alpha_i} \alpha_i \langle \boldsymbol{w}, \mathbf{v}_i \rangle^2} \tag{68}$$

$$\leq \mathcal{W}_A \sup_{\boldsymbol{w}:\|\boldsymbol{w}\|_{\boldsymbol{\Sigma_w}} \leq 1} \sqrt{\max_{i \in [d]} \frac{\lambda_i^{-1}}{\alpha_i} \sum_{i=1}^d \alpha_i \langle \boldsymbol{w}, \mathbf{v}_i \rangle^2}$$

$$= \mathcal{W}_A \sup_{\boldsymbol{w}:\|\boldsymbol{w}\|_A \leq 1} \sqrt{\max_{i \in [d]} \frac{\lambda_i^{-1}}{\alpha_i}} \|\boldsymbol{w}\|_{\boldsymbol{\Sigma_w}} = \mathcal{W}_A \sqrt{\max_{i \in [d]} \frac{\lambda_i^{-1}}{\alpha_i}}.$$

On the other hand, for $\boldsymbol{w} = \frac{1}{\sqrt{\alpha_j}} \mathbf{v}_j$ where $j \in \arg\max_{i \in [d]} \frac{\lambda_i^{-1}}{\alpha_i}$, it is $\|\boldsymbol{w}\|_{\boldsymbol{\Sigma_w}} = 1$ and also $\|\boldsymbol{w}\|_{\boldsymbol{\Sigma_\delta}^{-1}} = \sqrt{\max_{i \in [d]} \frac{\lambda_i^{-1}}{\alpha_i}}$, so the above bound is tight. $\qquad\square$

*Remark* C.4. Note that the previous result, and in particular inequality 68, implies that in order to most effectively control the robust error, we better select $\boldsymbol{\Sigma_w} = \boldsymbol{\Sigma_\delta}^{-1}$. To see this, consider a particular ground truth $\mathbf{w}^\star \in \mathbb{R}^d$, and let $\mathcal{H}$ be the smallest hypothesis class that contains hypothesis $\mathbf{w}^\star$ for a given geometry $\boldsymbol{\Sigma_w}$, that is: $\mathcal{H} = \{\mathbf{x} \mapsto \langle \mathbf{w}, \mathbf{x} \rangle : \|\mathbf{w}\|_{\boldsymbol{\Sigma_w}} \leq \|\|\mathbf{w}^\star\|_{\boldsymbol{\Sigma_w}}\|\}$. Then, from 68, we have:

$$\frac{\epsilon \|\mathbf{w}^\star\|_{\boldsymbol{\Sigma_w}}}{2\sqrt{n}} \sqrt{\max_{i \in [d]} \frac{1}{\lambda_i \alpha_i}} \geq \frac{\epsilon}{2\sqrt{n}} \|\mathbf{w}^\star\|_{\boldsymbol{\Sigma_\delta}^{-1}}, \tag{69}$$

where equality holds for all $\mathbf{w}^\star$ if and only if $\boldsymbol{\Sigma_w} \propto \boldsymbol{\Sigma_\delta}^{-1}$. In particular, if $\boldsymbol{\Sigma_w} = \boldsymbol{\Sigma_\delta}^{-1}$, then the above term becomes $\frac{\epsilon}{2\sqrt{n}} \|\mathbf{w}^\star\|_{\boldsymbol{\Sigma_\delta}^{-1}}$, and cannot be improved further without excluding $\mathbf{w}^\star$ from the class $\mathcal{H}$.

## C.1 Worst-case Rademacher complexity for $\ell_p$ norms

Awasthi et al. (2020) provides the following bound on the worst-case Rademacher complexity of linear hypothesis classes constrained in their $\ell_r$ norm.

**Theorem C.5.** *Theorem 4 in (Awasthi et al., 2020) Let $\epsilon > 0$ and $p, r \geq 1$. Define $\mathcal{H}_{\widetilde{r}} = \{\mathbf{x} \mapsto \langle \mathbf{w}, \mathbf{x} \rangle : \|\mathbf{w}\|_r \leq 1\}$. Then, it holds:*

$$\hat{\mathfrak{R}}_S(\widetilde{\mathcal{H}}_r) \leq \hat{\mathfrak{R}}_S(\mathcal{H}) + \epsilon \frac{\max(d^{1-\frac{1}{r}-\frac{1}{p}}, 1)}{2\sqrt{n}}. \tag{70}$$

The bound above suggests regularizing the weights in the $\ell_r$ norm, $r = \frac{p}{p-1}$, for effectively controlling the estimation error of the class.

# D  Parameter Exploration

This section presents the experimental details for all the figures in the main text and explore the model parameters in greater detail. For implementation details of our numerical procedures, please refer to Appendix E.

## D.1  Settings for Main Text Figures

All figures in the main text utilize the logistic loss function, defined as $g(x) = \log(1 + \exp(-x))$. Below, we detail the specific parameters for each figure.

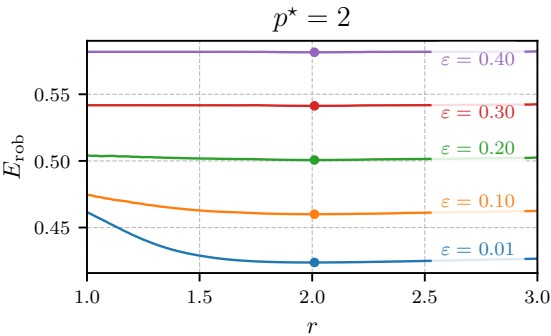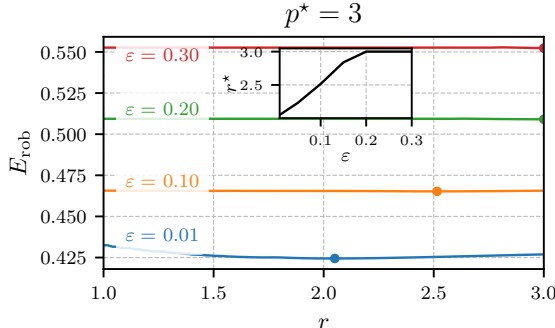

Figure 4: Robust error as a function of the regularization order $r$ for two different $p^\star$. By increasing the value of $\varepsilon$ we have that the optimal value $r^\star$ gets close to $p^\star$.

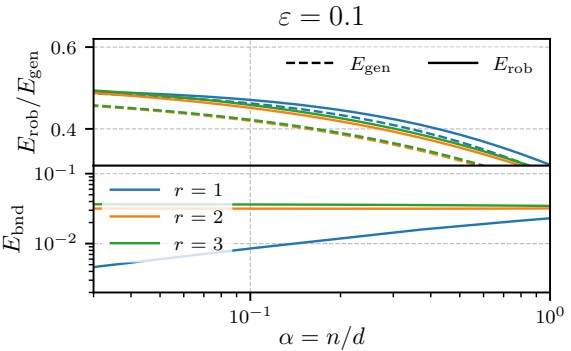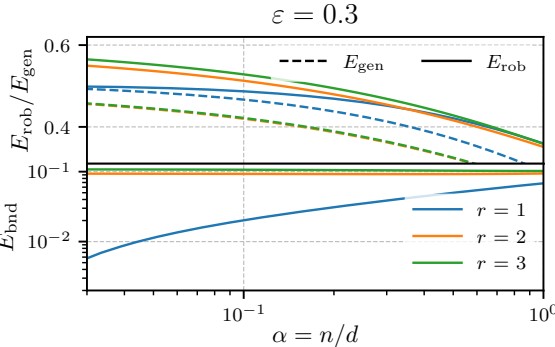

Figure 5: Robust error, generalization error and boundary error for different choices of regularization geometry $r$ as a function of the sample complexity $\alpha$. We see that the value of the errors increases with $\varepsilon$.

**Figure 1 (Left)** We optimize the regularization parameter $\lambda$ for each curve. Parameters: $\epsilon = 0.2$, noiseless regime ($\tau = 0$). Data points represent averages over 10 distinct data realizations with dimension $d = 1000$, varying sample size $n$ to adjust $\alpha$. Error bars indicate one standard deviation from the mean.

**Figure 2 (Right)** Generated in the noiseless case ($\tau = 0$) with optimal regularization parameter $\lambda$. We optimize robust error for regularizations $r = 2$ and $r = 1$ independently, then compute their difference.

**Figure 2 (Left)** We employ a Strong Weak Feature Model (SWFM) as defined in Tanner et al. (2025). This model implements a block structure on all covariances ($\boldsymbol{\Sigma_x}$, $\boldsymbol{\Sigma_\delta}$, $\boldsymbol{\Sigma_\theta}$, and $\boldsymbol{\Sigma_w}$), with block sizes relative to dimension $d$ denoted by $\phi_i$ for block $i$. We use two equal-sized feature blocks, totaling $d = 1000$. All matrices are block diagonal, with each block being diagonal. The values for each matrix are as follows

|  | $\boldsymbol{\Sigma_x}$ | $\boldsymbol{\Sigma_\delta}$ | $\boldsymbol{\Sigma_\theta}$ | Case $\boldsymbol{\Sigma_w} = \boldsymbol{\Sigma_\delta}$ | Case $\ell_2$ | Case $\boldsymbol{\Sigma_w} = \boldsymbol{\Sigma_\delta}^{-1}$ |
|---|---|---|---|---|---|---|
| First Block | 1 | 1 | 1 | 1 | 1 | 2.5 |
| Second Block | 1 | 2.5 | 1 | 2.5 | 1 | 1 |

All matrices are trace-normalized, with $\varepsilon$ values as specified in the figure. Again error bars indicate the deviation from the mean.

**Figure 2 (Right)** We optimize the regularization parameter $\lambda$ in the noiseless case ($\tau = 0$), with $\alpha = 1$. The inset is generated by conducting $r$ sweeps for 10 distinct $\varepsilon$ values. Each sweep comprises 50 points, with the minimum determined using `np.argmin`.

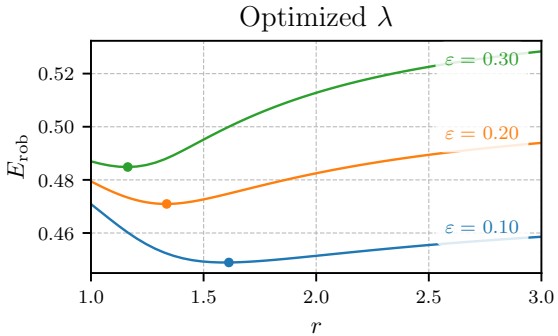
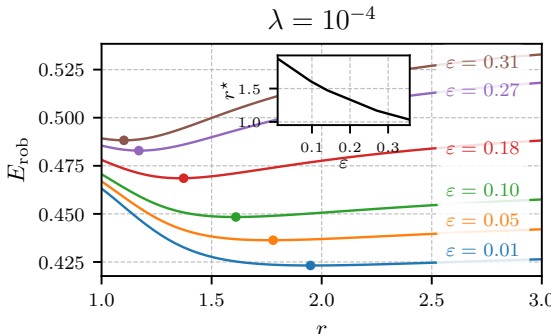

Figure 6: Robust generalization error as a function of regularization order $r$ for fixed versus optimized regularization strength $\lambda$. The comparison illustrates that the impact of $\lambda$ optimization does not change qualitatively the behavior of the optimal regularization geometry $r^\star$ as $\varepsilon$ increases.

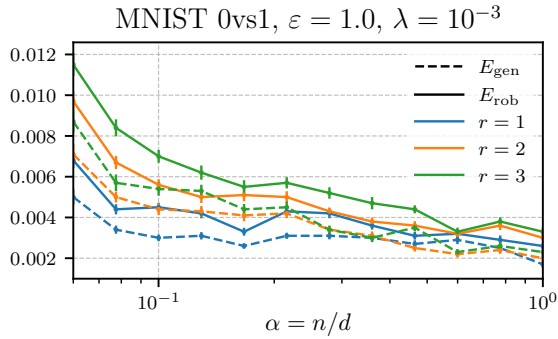
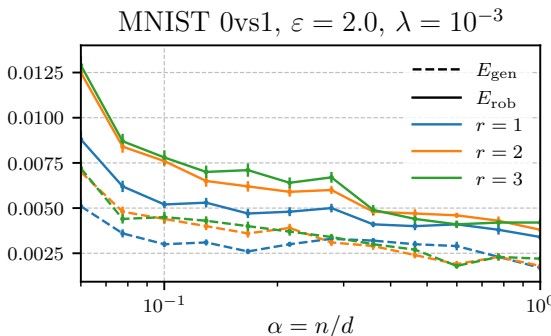

Figure 7: Generalization error and robust generalization error as a function of sample complexity $\alpha$ for MNIST binary classification (0 vs 1) with $\ell_\infty$ adversarial training. The optimal regularization is $r = 1$ (matching the attack norm), consistent with theoretical predictions from Section 5. Despite the non-Gaussian data distribution, the empirical results align closely with high-dimensional asymptotic theory.

## D.2 Additional Parameter Exploration and Dataset Choice

We now present some additional exploration of the model in some different regimes.

**Figure 4** These figures display theoretical results for attack perturbations constrained by $\ell_2$ (Left) and $\ell_{3/2}$ (Right) norms. We vary $\varepsilon$ as shown and use the noiseless regime ($\tau = 0$). Parameters: $\alpha = 0.1$, optimal $\lambda$. Each sweep comprises 50 points, with minima determined using `np.argmin`. Points on the curves indicate the minimum for each $\varepsilon$ value.

**Figure 5** This figure illustrates generalization metrics as a function of $\alpha$ for various regularization geometries. We present results for two attack strengths: $\varepsilon = 0.1$ (Left) and $\varepsilon = 0.3$ (Right). Both use optimal $\lambda$ values. This figure can be compared to Figure 1 (Left).

**Figure 6** Both panels show robust generalization error versus regularization geometry $r$, with $\alpha = 0.1$. The right panel optimizes regularization strength $\lambda$, while the left uses a fixed value $\lambda = 10^{-4}$.

**Figure 7** Empirical validation on MNIST binary classification (digits 0 vs 1). We train a linear classifier via $\ell_\infty$ adversarial training eqs. (6) and (7), where MNIST images are flattened to $\boldsymbol{x}_\mu \in \mathbb{R}^{784}$ and normalized element-wise to $[0, 1]$ (*i.e.* $\boldsymbol{x}_\mu \mapsto \boldsymbol{x}_\mu/255$). Labels are mapped to $y_\mu \in \{\pm 1\}$. The ridge parameter is fixed at $\lambda = 10^{-3}$ across all experiments. We evaluate the population adversarial error eq. (2) and standard generalization error eq. (3) via the empirical version of the expectation with $n_{\text{test}} = 1000$ samples from the test set. Results are shown as a function of the sample complexity

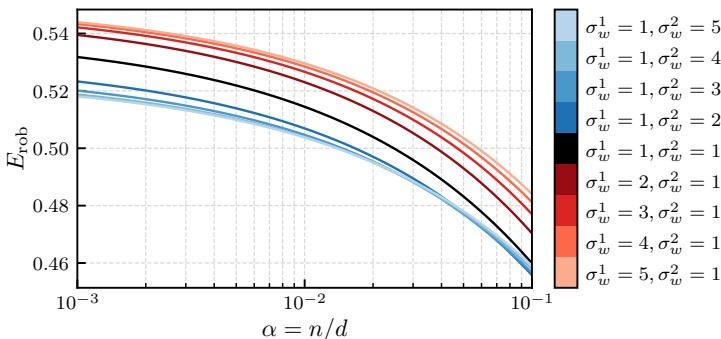

Figure 8: Robust generalization error as a function of sample complexity $\alpha$ for the Mahalanobis model defined in Section 3.2 and a sweep in different regularization matrices. We see that interpolating from the value of $\mathbf{\Sigma_w} = \mathbf{\Sigma_\delta}$ to $\mathbf{\Sigma_w} = \mathbf{\Sigma_\delta^{-1}}$ decreases the value of the error.

ratio $\alpha = n/d$ where $d = 784$ is kept fixed. Two attack budgets are considered: $\varepsilon = 1.0$ (Left) and $\varepsilon = 2.0$ (Right). We compare different regularization norms $r \in \{1, 2, \infty\}$ to identify the optimal choice matching the attack geometry.

**Figure 8** We study the behaviour of the robust generalization error for more choices of regularization matrix $\mathbf{\Sigma_w}$ for the model defined in Section 3.2. This model implements a block structure on all covariances ($\mathbf{\Sigma_x}$, $\mathbf{\Sigma_\delta}$, $\mathbf{\Sigma_\theta}$, and $\mathbf{\Sigma_w}$). We use two equal-sized feature blocks. All matrices are block diagonal, with each block being diagonal. The values for each matrix are as follows

|  | $\mathbf{\Sigma_x}$ | $\mathbf{\Sigma_\delta}$ | $\mathbf{\Sigma_\theta}$ | $\mathbf{\Sigma_w}$ |
|---|---|---|---|---|
| First Block | 1 | 1 | 1 | $\sigma_w^1$ |
| Second Block | 1 | 5 | 1 | $\sigma_w^2$ |

All matrices are trace-normalized, with $\varepsilon$ values as specified in the figure. We vary the values of $\sigma_w^1$ and $\sigma_w^2$ and keep $\lambda = 10^{-3}$, $\varepsilon = 0.1$ fixed.

## E  Numerical Details

The self-consistent equations from Theorems 3.7 and 3.16 are written in a way amenable to be solved via fixed-point iteration. Starting from a random initialization, we iterate through both the hat and non-hat variable equations until the maximum absolute difference between the order parameters in two successive iterations falls below a tolerance of $10^{-5}$.

To speed-up convergence we use a damping scheme, updating each order parameter at iteration $i$, designated as $x_i$, using $x_i := x_i \mu + x_{i-1}(1 - \mu)$, with $\mu$ as the damping parameter.

Once convergence is achieved for fixed $\lambda$, hyper-parameters are optimized using a gradient-free numerical minimization procedure for a one dimensional minimization.

For each iteration, we evaluate the proximal operator numerically using SciPy's (Virtanen et al., 2020) Brent's algorithm for root finding (`scipy.optimize.minimize_scalar`). The numerical integration is handled with SciPy's quad method (`scipy.integrate.quad`), which provides adaptive quadrature of a given function over a specified interval. These numerical techniques allow us to evaluate the equations and perform the necessary integrations with the desired accuracy.

Regarding the computer hardware all the experiments have been run on consumer grade hardware, specifically MacStudio M2 Ultra 2022, and none of the run took more than 1 day of CPU time.

