# OpenReview forum: "On the Geometry of Regularization in Adversarial Training: High-Dimensional Asymptotics and Generalization Bounds"
_TMLR — Accepted by TMLR_

### Review · Reviewer_rjdr · 2025-12-11

**Summary Of Contributions:**

**Summary**

This paper provides an analysis of how different regularization schemes in adversarial training impact both standard and robust generalization. Specifically, for model selection via regularization in Robust Empirical Risk Minimization (RERM), this work investigates how the geometries of the regularization (defined by the norm $||\cdot||$) and admissible perturbations interact to determine the robust generalization error.

The theoretical analysis is done in the high-dimensional proportional limit, i.e., when both the input dimension ($d$) and the number of training samples ($n$) diverge to infinity while maintaining a fixed ratio $\alpha = n/d$. Within this regime, the authors focus on binary classification using generalized linear models. They get an exact asymptotic characterization of the RERM estimator in two settings: isotropic gaussian input and $l_p$ norm geometry, and correlated gaussian input and Mahalanobis norm, with the latter considered under simultaneous diagonalizability. These high-dimensional characterizations are then reduced to low-dimensional summary statistics, determined as solutions to a set of self-consistent equations, which are solved via an iteration scheme and validated against numerical evidence. Additionally, the authors complement this asymptotic analysis by establishing uniform convergence bounds based on Rademacher Complexity. Lastly, both approaches suggest an insight that in certain cases choosing a regularization norm that corresponds to the dual norm of the perturbation (e.g., using $l_1$ regularization to defend against $l_\infty$ attacks) could be beneficial.




**Strengths**
- *Sound Theoretical Foundation*: The work has a rigorous theoretical analysis .
- *Tractable Characterization*: The authors provide a characterization of the problem, which is complemented by a numerically tractable procedure that gives predictions aligning with empirical results.
- *Extension to Structured Covariates*: The work explores beyond isotropic Gaussian distributions.
- *Novel Insights*: The theoretical results suggest interesting insights, i.e., suggesting benefits of using one regularization norm over the other depending on the perturbation geometry, attack budget and sample complexity.

**Weakness**

- *Gaussian Assumption*: The characterized asymptotics rely strictly on Gaussian covariates. The authors assume, but do not prove, that these results transfer to real-world distributions via universality, leaving a gap between theory and practice .
- *Model simplicity*: The analysis is restricted to linear binary classification. This simplifies the complex, non-convex optimization landscape of the deep neural networks that practitioners actually use for adversarial training.
- *Qualitative bound*: The derived Rademacher complexity bounds are acknowledged to be numerically loose. They function as a qualitative confirmation of the "dual norm" principle rather than offering tight, quantitative guarantees.

**Additional Comments:**

**Questions**
- Can some theoretical insights be gathered from the summary statistics formulas obtained in Theorem 3.7. And Theorem 3.15. without running the numerical iteration scheme. If not, what would be a roadblock?
- On page 8, below Corollary 4.4 can you give a precise mathematical statement from which we can deduce that choosing $\Sigma_{\delta}^{-1}$ can more effectively control the robust generalization error?
- What would be required to move away from the simultaneous diagonalizability assumption used in Section 3.2.?
- What is the main technical difference in the proofs of the results in this work in section 3.1. and 3.2. and (Tanner et. al. 2025)?


Tanner, Kasimir, Matteo Vilucchio, Bruno Loureiro, and Florent Krzakala. "*A High Dimensional Statistical Model for Adversarial Training: Geometry and Trade-Offs.*" In The 28th International Conference on Artificial Intelligence and Statistics.

**Audience:**

Yes

**Audience Explanation:**

Yes, I believe that these findings can be of interest to the TMLR audience. In general, this work tackles a relevant question, as adversarial robustness remains an unsolved challenge in modern machine learning. This work addresses the need for simple, principled guidelines on how design choices, specifically regularization, affect statistical efficiency in robust learning. Moreover, the precise asymptotic characterization in the high dimensional limit as well as Rademacher uniform convergence bound can be valuable to the learning theory community. This is supported by the fact that this work offers a complementary theoretical picture to recent work by (Tanner et. al. 2025) by extending their analysis of structured covariates to Mahalonobis regularization norms. Lastly, even though these findings are made for linear models, they could still inform model selection in practice. For example the finding in Figure 1 (Right) that optimal regularization transitions smoothly (e.g., from $l_2$ to $l_1$) as the attack budget $\epsilon$ grows offers a specific, testable insight for hyperparameter tuning. This could provide a geometric justification for choosing sparsity-inducing norms in high-perturbation regimes.

**Broader Impact Concerns:**

I have no broader impact concern.

**Claims And Evidence:**

Yes

**Claims Explanation:**

For the most part the claims made in the submission are supported by accurate convincing and clear evidence. The authors accurately support the first two main contributions outlined in the introduction, i.e., the exact asymptotic characterizations of the standard and robust generalization error, and uniform convergence bound. However, I have a reservation regarding Contribution 3, which claims that regularizing with the dual norm can yield "**significant** benefits". Even though this claim is technically true, in order to have any practical implications I believe that presented evidence to support it should be more robust. For example, while Figure 2 (Left) does show the dual norm ($\Sigma_\delta^{-1}$) outperforms both the perturbation norm ($\Sigma_\delta$) and the standard Euclidean norm ($l_2$), the empirical evidence is somewhat limited to this specific matrix setup. To fully support such a claim, it would be beneficial to see comparisons across a wider variety of matrix structures or regularization geometries beyond these specific cases. Furthermore, while the authors use Rademacher complexity bounds to theoretically motivate the dual norm choice, they acknowledge these bounds are not numerically tight, meaning they should be viewed as qualitative intuition rather than definitive proof of the claim.

**Requested Changes:**

**Proposed changes** (would help supporting recommendation of acceptance)
- It would be good to see more evidence to support the claim that dual norm of the perturbation can yield **significant** benefits, or to make the claim weaker. The former can be done by e.g. analyzing a wider variety of matrix structures or regularization geometries beyond analyzed specific cases.
-  Give support to the claim that optimizing the robust empirical risk as defined in (6) is a widely adopted approach.

**Typos encountered & notation change suggestions** (none of them are critical for recommendation of acceptance):
- Page 3 (top): $z$ is used, but there is no definition of what z is. Even though it is clear from the context, I think it should be defined.
- Pages 3 (bot), 19 (bottom): Both $r$ and $\tilde{r}$ show up throughout the paper. I think this is a typo, and only $\tilde{r}$ should be used.
- Pages 4,5, etc.: Throughout the paper equation (6) is referred to as the definition of the RERM problem, whereas (6) is the definition of the empirical regularized risk and (7) is closer to the actual definition of the RERM problem. Therefore, I think that when referring to the RERM problem, one should refer to the equation (7), or if referring to equation (6) to be more precise.
- Pages 5 and 6 (mid): Make precise what are the variables from the 8 equations that the solution $(m*,q*,P^*)$ should correspond to, as it is not uniquely determined just from the statement.
- Pages 5 and 6 (mid): Specify what type of convergences are used in Theorems 3.5 and 3.14.
- Page 6 (top): I think it's not clear what perturbations under a Mahalanobis norm mean, since the subscript in $\Sigma_\delta$ is $\delta$ which is used as a variable. This suggests that $\Sigma_\delta$ also changes with $\delta$. I suppose this should not be the case, and thus It would be worth it to clear up this ambiguity.
- Page 6 (mid): The notation of $\omega_i$ and $w_i$ can be confused. I would propose to use something else instead of $\omega_i$.
- Pages 8 and 24-26 (mid): I think $m$ is a typo and should be substituted by $n$.
- Pages 8 and 24-26: There is matrix $A$ that shows up in notation that probably should be matrix $\Sigma_{w}$.
- Pages 24-26: As the statements of propositions in Appendix C are the same as in the main body,  it would be good to acknowledge that they are indeed just restatements.
- Page 26 (top): I think $\Sigma^{-1}$ should be $\Sigma_{\delta}^{-1}$.

---

> ### Author Response · Authors · 2026-02-02
>
> Thank you for your review and your help in strengthening our contribution.
>
> > It would be good to see more evidence to support the claim that dual norm of the perturbation can yield significant benefits, or to make the claim weaker. The former can be done by e.g. analyzing a wider variety of matrix structures or regularization geometries beyond analyzed specific cases.
>
> We appreciate this suggestion. To strengthen the empirical support, we plan to add simulations on the MNIST 0 vs. 1 classification task that exhibit the same qualitative phenomenon (see our response to Reviewer jnAD).
> We will also explored additional choices regularization matrices in Appendix D, interpolating between $\Sigma_w = \Sigma_\delta$ to $\Sigma_w = \Sigma_\delta^{-1}$.
> We additionally changed the phrasing in the introduction to better match the style described by the reviewer.
>
> > Give support to the claim that optimizing the robust empirical risk as defined in (6) is a widely adopted approach.
>
> This is the standard approach of (regularized) adversarial training. Eq. (6) simply specializes it in the linear case.
>
> *Questions*
>
> > Can some theoretical insights be gathered from the summary statistics formulas obtained in Theorem 3.7. And Theorem 3.15. without running the numerical iteration scheme. If not, what would be a roadblock?
>
> The system of equations in the theorem could provide insights into specific questions. For instance, studying them perturbatively might help reveal how the generalization error scales with model parameters. As an example, Aubin et al. (2020) derived such scalings for a robust regression setting.
>
> However, the analytical progress in that work relies critically on the choice of loss function ($\ell_2$ loss), which admits closed-form proximal operators, as well as on the regression setting they consider. In our binary classification context, the proximal operator does not have an analytical form for commonly used classification losses, making similar closed-form derivations significantly more challenging to obtain.
>
> > On page 8, below Corollary 4.4 can you give a precise mathematical statement from which we can deduce that choosing  can more effectively control the robust generalization error?
>
> We added a clarification in Remark C.4 in Appendix C on how the claim follows from Corollary 4.4. We also added a pointer to this in the main text.
>
> > What would be required to move away from the simultaneous diagonalizability assumption used in Section 3.2.?
>
> This is an assumption which will require a more stingent knownledge of the limiting distribution of the matrices considered.
> Specifically to properly define the high dimensional limit one should have a characterization of how the different eigenvectors (and their dot product) of all related matrices behave in the limit. One could make several assumtions on this, but we do not believe that there exist natural enough assumptions that can provide signficant insight into this kind of systems.
>
> We also note that the simultaneous diagonalizability assumption is less restrictive than what it appears. In fact, it is equivalent to the matrices having common principal components.
> This is a very common modelling assumption in the literature of Signal Processing, Bayesian Statistics and more, as input correlations and measurement noise, for example, frequently share a common structure. This is the same underlying assumption that supports PCA for data analysis.
>
> We will add a discussion of this behaviour in the revised version.
>
> > What is the main technical difference in the proofs of the results in this work in section 3.1. and 3.2. and (Tanner et. al. 2025)?
>
> In a nutshell, Tanner et al. 2024 study Mahalanobis-norm perturbations and regularization with an $\ell_2$ penalty, while Section 3 in our paper presents results for (a) Mahalanobis-norm perturbations and general Mahalanobis-norm regularizations and (b) $\ell_p$ norms for both perturbation and regularization. On the technical side the proofs that appear in [Tanner et al. 2024] are based on a mapping to a Generalised Approximate Message Passing algorithm. Our results are proven via Gordon's Theorem (Theorem A.1) and allow for a broader range of loss and regularization functions.

---

### Review · Reviewer_513z · 2025-12-29

**Summary Of Contributions:**

The paper presents a theoretical analysis of the statistical learning error of adversarial empirical risk minimization for high-dimensional linear classification problems. The authors mainly consider robust empirical risk minimization (RERM) under the proportional limit regime. They first derive the PDEs satisfied by the exact asymptotics under both the $L_p$ norm and the Mahalanobis norm. Then, using Rademacher complexity, they derive analytic formulas for upper bounds on the generalization error. Based on these upper bounds, they propose using the inverse of the variance of the predictor in the Mahalanobis norm. They further provide experiments to justify the theoretical claims.

**Audience:**

Yes

**Audience Explanation:**

On the theory side, it gives an attempt to derive the exact asymptotic characterization of the RERM across several attack and regularization norms, together with a complementary uniform-convergence/Rademacher-complexity analysis. Although the current results do not fully close the story, the framework feels like a clean base for researchers who want to push this direction further. Overall, it is written clearly and reads as an incremental but useful contribution that can support future extensions and discussion.

**Claims And Evidence:**

Yes

**Claims Explanation:**

The theoretical results are presented clearly without significant issues. All bounds are stated with explicit coefficient factors, making the claims trustworthy. The experimental settings are well designed, with complete details provided in the appendix.

**Requested Changes:**

I suggest that the authors reconsider the role of the exact asymptotics results in the paper. As they stand, the results are incomplete, and most of the intuitive insights of choice of regularizations come from the Rademacher-complexity–based upper bounds. The authors should rethink and position the exact asymptotic analysis in a more appropriate way. At present, these results appear distracting from the main story. It may be worth adding more discussion of the Rademacher-complexity bounds or including additional experimental settings. Alternatively, is it possible that the exact asymptotics provide additional information that would be missed if one focuses only on Rademacher complexity?

---

> ### Author Response · Authors · 2026-02-02
>
> Thank you very much for your encouraging comments and your review!
>
> > I suggest that the authors reconsider the role of the exact asymptotics results in the paper. As they stand, the results are incomplete, and most of the intuitive insights of choice of regularizations come from the Rademacher-complexity–based upper bounds. The authors should rethink and position the exact asymptotic analysis in a more appropriate way. At present, these results appear distracting from the main story. It may be worth adding more discussion of the Rademacher-complexity bounds or including additional experimental settings. Alternatively, is it possible that the exact asymptotics provide additional information that would be missed if one focuses only on Rademacher complexity?
>
> Regarding additional experiments, we kindly point you to our response to Reviewer jnAD (and Appendix D.4 of the revised version) for some additional simulations with realistic data.
>
> Regarding additional intuition from the exact asymptotic results, we would like to point you to our fine-grained analysis in Appendix B.8. There, we perform a low-order expansion of the high-dimensional description to study the behavior of the overlaps in the low $\alpha$ regime (low sample complexity). We find different scalings for the overlap between the true solution $w^\star$ and the estimated one $\hat{w}$, depending on the choice of regularization. This shows how high-dimensional analysis can be used to provide complementary predictions to uniform convergence bounds. We will add a pointer to this section after the presentation of our asymptotic results in the main text.

---

### Review · Reviewer_jnAD · 2025-12-31

**Summary Of Contributions:**

This manuscript studies the binary classification problem under Robust ERM, with a particular focus on the effect of regularization. The authors consider two approaches: (i) under Gaussian covariates and a generative assumption, they derive precise standard and robust generalization risk curves for $L_p$ and Mahalanobis norm perturbations; (ii) they derive robust generalization bounds using uniform convergence argument and Rademacher complexity analysis for different norm geometries. They also conduct experiments to validate their theoretical results.

**Strengths:**

* The precise derivations in the Gaussian setting provide insight into the effect of regularization, which may generalize to a broader class of distributions via universality. In this sense, the results are interesting.

**Weaknesses:**

* Although the use of Gaussian data is justified in the paper through universality results from the literature, I could not find experiments beyond the Gaussian case for the asymptotic risk (please let me know if I am missing something). The authors could study more general sub-Gaussian covariates to empirically assess when universality holds or fails.
* The precise results, as they stand in the manuscript, are difficult to interpret: the effect of the regularization norm and strength is not clearly visible, and this aspect is not analyzed by the authors either. For example, For example, can the behavior observed in Figure 2 (right) be derived from the self-consistent equations?
* The precise asymptotic analysis and the Rademacher complexity–based bounds feel more complementary than connected. What is the conceptual connection between these two sections? For example, can we show (or disprove) the tightness of the uniform convergence bound through the precise asymptotic analysis?

**Additional Comments:**

Some minor comments:
* Page 5, third line from the end: convariates -> covariates.
* Page 8, second line: Proposition -> proposition.

*As a final question*, what happens if the perturbations at training and test time are different? Does the analysis in the paper provide insight into how a mismatch between perturbations affects robust generalization? I would appreciate further comments on this.

**Audience:**

Yes

**Audience Explanation:**

See the "Strengths" part above

**Claims And Evidence:**

Yes

**Claims Explanation:**

The paper is primarily theoretical, and the theorems are consistent with the experimental results. I partially reviewed the proofs, and the results appear to be correct.

**Requested Changes:**

Please address the Weaknesses part above.

---

> ### Author Response · Authors · 2026-02-02
>
> Thank you very much for your review! We respond to your comments:
>
> > Although the use of Gaussian data is justified in the paper through universality results from the literature, I could not find experiments beyond the Gaussian case for the asymptotic risk (please let me know if I am missing something). The authors could study more general sub-Gaussian covariates to empirically assess when universality holds or fails.
>
> Thank you for the suggestion. On the theoretical side, a growing body of work on universality suggests that many asymptotic predictions extend to broader classes of distributions, including with sub-Gaussian covariates (e.g., Montanari & Saeed, 2022; Dandi et al., 2023). A formal extension would require replacing Gaussian tools with more general concentration and invariance arguments. However, maintaining our level of generality - most notably, allowing arbitrary non-increasing convex loss functions - would require substantial additional technical machinery, which we view as outside the scope of this initial contribution. We consider this a promising direction for future work.
>
> To address your request for experiments beyond the Gaussian case, we include some additional simulations, which we now describe.
> We consider the case of 0 vs 1 MNIST classification. We take 0 and 1 MNIST images, normalize their pixel values in $[0,1]$  and flatten them to have a $d=784$ dimensional $\boldsymbol{x}$. The labels are converted to be $\pm 1$. We then train as per eqs. (6,7) with this dataset, using subsets of a suitable size to fix various $\alpha = n/d$ and then test robust and clean error as per eqs. (3,4) with $1000$ new samples.
> We present value of the robust generalization error $E_{rob}$ for different values of $n$ the results of two experiments with $\varepsilon = 1.0, 2.0$ in Appendix D.2 Figure 7 of the revised version.
>
> While the behaviour is less "smooth" we still see that by reducing the number of training samples (lowering $n$ and thus $\alpha$) the robust error differs based on the choice of the regularization. The optimal choice of regularization is $r=1$, as expected from our analysis in Section 3. Furthermore, the high-dimensional predictions closely match the simulations, suggesting that our theoretical assumptions regarding a Gaussian distribution might not be limiting for realistic data. In the camera ready version, we will include a discussion of these results.
>
> > The precise results, as they stand in the manuscript, are difficult to interpret: the effect of the regularization norm and strength is not clearly visible, and this aspect is not analyzed by the authors either. For example, For example, can the behavior observed in Figure 2 (right) be derived from the self-consistent equations?
>
> We want to state that the results in Figure 2 (right) are obtained by the numerical solution of the system of self consistent equations.
> While the full behaviour shown in the Figure is not possible to be characterized analytically (in a simple and intuitive form) we gain some insights in why the specific behaviour is the one shown in Appendix B.8.
> What we show there is that, by expanding the self consistent equations in the low $\alpha$ regime, we cleary distinguish the behaviours of $r=1$ and $r=2$ (Figure 3). We will add a pointer to this section after the presentation of our asymptotic results in the main text.
>
> > The precise asymptotic analysis and the Rademacher complexity–based bounds feel more complementary than connected. What is the conceptual connection between these two sections? For example, can we show (or disprove) the tightness of the uniform convergence bound through the precise asymptotic analysis?
>
> One could perhaps study uniform convergence bounds in specific distributional cases (such as the Gaussian case), but these bounds are typically loose, and there is little reason to invoke high-dimensional analysis to establish this. However, their competitive advantage lies in the fact that they can motivate certain algorithmic interventions (such as choosing the dual regularization norm as we saw in the paper), despite not tightly bounding the actual generalization gap. Indeed, the two approaches are complementary, and one conceptual contribution of our work is to show that they can inform different aspects of model selection (or regularization).
>
> > As a final question, what happens if the perturbations at training and test time are different? Does the analysis in the paper provide insight into how a mismatch between perturbations affects robust generalization? I would appreciate further comments on this.
>
> This is a very interesting question, but we believe it lies outside the scope of the paper. We would expect the quantitative insights to hold when modifying the perturbation **strength** at test time, but it is unclear what would happen if the threat model (i.e., the perturbation geometry/ball) also shifts at test time.

---

### Decision · Action_Editor_6GGu · 2026-03-11

**Recommendation:** Accept as is

**Audience:**

Yes

**Audience Explanation:**

The focus of the paper is on the theory of adversarial robustness through the lens of high dimensional statistics and regularization. This is of interest to the learning theory community, hence relevant to TMLR.

**Claims And Evidence:**

Yes

**Claims Explanation:**

The paper is of theoretical nature and the claims appear to be correct and backed by valid numerical experiments.